



# Parameter Sensitivity Study of Energy Transfer Between
# Mesoscale Eddies and Wind-Induced Near-Inertial Oscillations
Yu Zhang[1], Jintao Gu[1], Shengli Chen[1*], Jianyu Hu[2], Jinyu Sheng[3], Jiuxing Xing[1]
[1]Institute for Ocean Engineering, Shenzhen International Graduate School, Tsinghua University, Shenzhen, 518055,
China
[2]State Key Laboratory of Marine Environmental Science, College of Ocean and Earth Sciences, Xiamen University,
Xiamen, 361102, China
[3]Department of Oceanography, Dalhousie University, Halifax, NS B3H 4R2, Canada
*Correspondence to*: Shengli Chen (shenglichen@sz.tsinghua.edu.cn)
**Abstract.** Analyses of current observations and numerical simulations at two moorings in the
northern South China Sea reveal the transfer of near-inertial energy between the background
currents associated with mesoscale eddies and near-inertial currents (NICs). A series of numerical
experiments are conducted to determine important parameters affecting the energy transfer
between idealized mesoscale eddies and NICs generated by rotating winds. Speeds of NICs
transferred by both cyclonic and anticyclonic mesoscale eddies increase linearly with the wind
stress and eddy strength. The transferred NICs in anticyclonic eddies have current amplitudes of
about six times larger than in cyclonic eddies. The translation speed of the mesoscale eddy and the
wind rotation frequency also affect the conversion of NICs. The energy transfer rate is elevated
with the increase of the positive Okubo-Weiss parameter. A simple theoretical analysis is
conducted to verify our findings based on numerical results. Analytical solutions confirm the
evident asymmetry of the energy transfer between anticyclonic and cyclonic eddies, and
demonstrate quantitatively the relationship between the wind stress and the near-inertial energy
transferred by mesoscale eddies.
## 1 Introduction
Near-inertial oscillations (NIOs) are very common in the global ocean and they appear as a
prominent peak in the spectrum of ocean currents (Garrett, 2001). NIOs contain almost half the
total kinetic energy of the internal waves and significantly contribute to the vertical shear in the
internal waveband (Ferrari & Wunsch, 2009). When surface winds with high spatiotemporal




variations act at the ocean surface, strong NIOs could occur in the ocean surface mixed layer (SML,
Chen et al., 2015a; D'Asaro et al. 1995; Pollard & Millard, 1970). At the base of the SML, near-
inertial internal waves (NIWs) are generated through the horizontal convergence and divergence
of the SML (Gill, 1984). These NIWs are free to radiate to the thermocline and deep waters, and
low-mode NIWs with long wavelengths can propagate at least hundreds of kilometers toward the
equator from their source regions (Alford, 2003; Jochum et al., 2013; Munk & Wunsch, 1998).
NIOs not only affect the energy, momentum, and material transport in the upper ocean, but also
play an important role in maintaining diapycnal mixing and global ocean circulation (Chen et al.,
2016; Greatbatch, 1984; Price et al., 1986; Wunsch & Ferrari, 2004).

Due to the turbulent and inhomogeneous nature of the ocean, the central frequency of NIOs

is influenced by the $\beta$ effect and shows a significant blue or red shift induced by the relative
vorticity of mesoscale eddies (Chen et al., 2015b; Elipot et al., 2010; Kunze, 1985; Mooers, 1975;
Perkins, 1976; Sun et al., 2011). If the magnitude of the gradient in the relative vorticity is larger
than the $\beta$ effect (Chelton et al. 2011), mesoscale eddies also modulate the energy distribution and
propagation of the near-inertial motions (van Meurs, 1998; Wang et al., 2024). Previous studies of
current observations in the northwestern South China Sea (nSCS) demonstrated that the near-
inertial energy propagates both upwards and downwards under the influence of the anticyclonic
eddies (Chen et al., 2013; Zhai et al., 2007). Using the in situ observations and ray-tracing
techniques, Jaimes and Shay (2010) demonstrated that anticyclonic eddies trap the near-inertial
kinetic energy which rapidly propagates vertically below the thermocline and even to the deep
ocean. Young and Jelloul (1997) suggested that the anticyclonic eddies can improve the vertical
propagation rate of the near-inertial energy by deepening the thermocline. Zhai et al. (2005)
demonstrated that the wind generated near-inertial kinetic energy is also high in the strong
mesoscale motion regions.

Mesoscale eddies and NIOs are energetic in the SML (Bühler & McIntyre, 2005; Vanneste,

2013; Xie & Vanneste, 2015). Mesoscale eddies not only change the spatial distribution of NIOs,
but also exchange the energy with NIOs through the nonlinear interaction (Muller, 1976; Thomas,
2012). Based on the observational studies of specific NIO events, Noh and Nam (2020) found that
mesoscale eddies can effectively enhance the intensity of NIOs. The energy transfer also occurs
between NIOs and low-frequency geostrophic currents through the nonlinear interaction (Liu et
al., 2023; Thomas, 2012; Whalen et al., 2020). Jing et al. (2018) demonstrated that the large-scale



geostrophic currents of the Gulf Stream affect the distribution of near-inertial energy. Whitt and
Thomas (2015) suggested that, in a unidirectional laterally sheared geostrophic flow, a continuous
energy transfer occurs between mesoscale eddies and NIOs. In the Kuroshio extension, due to the
change of the effective Coriolis frequency caused by the relative vorticity of mesoscale eddies, the
energy exchange efficiency between the anticyclonic eddies and NIOs is about twice that between
the cyclonic eddies and NIOs (Jing et al., 2017). Based on numerical results in the Icelandic Basin,
Barkan et al. (2021) found that a significant energy transfer occurs between NIOs and sub-inertial
motions, with the energy transfer rate in winter and summer are about half and a quarter of the
local near-inertial wind energy input, respectively. The above and other studies suggested that the
energy transfer processes between mesoscale eddies and NIOs should play an important role in the
ocean energy cascade (Alford et al., 2016; Ford et al., 2000; McWilliams, 2016; Thomas, 2017).
Nevertheless, most of previous studies focused on the energy transfer rate and efficiency. There is
a knowledge gap in the amplitude of the near-inertial energy transferred by the mesoscale eddy
and the sensitivity of the above-mentioned energy transfer to mesoscale eddies and wind
parameters. The main objective of this study is to quantity the energy transfer between the
mesoscale eddies and wind-induced NIOs.

This paper is structured as follows. Section 2 provides a brief description of observational

data and reanalysis used in this study. Section 3 presents the original and modified slab models.
Model results for the energy transfer between mesoscale eddies and the NIOs are given in Sect. 4.
A series of sensitivity experiments for determining the important factors affecting the energy
transfer is presented in Sect. 5. The results of sensitivity experiments are verified through
theoretical analysis in Sect. 6. Summary and discussions are given in Sect. 7.
**2 Observational and Reanalysis Data**

Current observations at two subsurface moorings named S2 and S3 respectively in the nSCS

(marked in Fig. 1) are analyzed here. The current observations at these two moorings were made
using the Acoustic Doppler Current Profilers (ADCPs). Mooring S2 is located at 117°39.619′ E
and 21°37.001′ N. Current observations at this location were made at depth bins from 58 m to 442
m from 22 August 2016 to 8 May 2017. At ADCP mooring S2, the vertical sampling interval is
16 m and the time internal is 60 minutes. ADCP mooring S3 is located at 117°26.528′ E and
21°52.945′ N. Current observations at this mooring were made at depth bins from 37 m to 229 m





during the same observational period as at location S2. At mooring S3, the vertical sampling
interval is 8 m and the time internal is 30 minutes.
Hourly winds at 10 m above the mean sea level in the nSCS were extracted from the European
Centre for Medium-Range Weather Forecasts (ECMWF) ERA5 reanalysis. The ERA5 winds from
July 2016 to June 2017 with an interval of 1 hour and the horizontal resolution of 0.5° are used to
calculate the hourly wind stress to be used as the model forcing.
The surface geostrophic currents used in this study were derived from ECMWF, using the sea
level grid data inferred from the global marine satellite observations provided by the Copernicus
Atmosphere Monitoring Service (C3S). The sea surface height anomaly and geostrophic current
data in the nSCS from July 2016 to June 2017 are used here, which have a horizontal resolution of
0.25° and a time interval of 24 hours.
The mixed layer depth (MLD) for the nSCS was extracted from the 2018 edition of the World
Ocean Atlas (WOA2018) ([www.ncei.noaa.gov/products/world-ocean-atlas](www.ncei.noaa.gov/products/world-ocean-atlas)), with a horizontal
resolution of 0.25°. The MLD at each location is defined as the depth at which the vertical change
of the potential density from the ocean surface is 0.125 (sigma units).



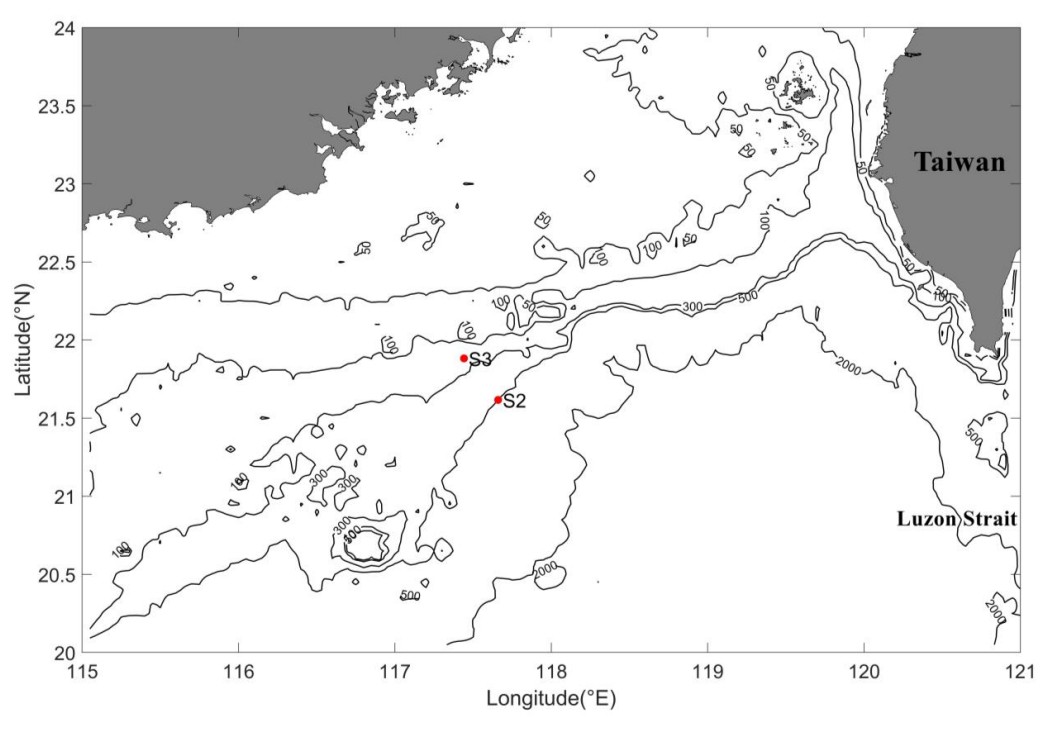

**Figure 1.** Major bathymetric feature of the northern South China Sea. Two red dots represent locations of two ADCP moorings named S2 and S3. Black contours represent isobaths in meter.

## 3 Method

### 3.1 Strain and Vorticity of a Mesoscale Field

The vertical component of the relative vorticity ($\zeta$) has been used to measure the rate of fluid rotation within a mesoscale eddy, which is defined as

$$\zeta = \frac{\partial V}{\partial x} - \frac{\partial U}{\partial y}, \tag{1}$$

where $U$ and $V$ are zonal (eastward) and meridional (northward) surface geostrophic currents, respectively.

The normal and shear components of the rate of strain tensor, $S_n$ and $S_s$ are defined as



$$S_n = \frac{\partial U}{\partial x} - \frac{\partial V}{\partial y}, \tag{2}$$

$$S_s = \frac{\partial V}{\partial x} + \frac{\partial U}{\partial y}. \tag{3}$$

The relative importance of total strain and relative vorticity is diagnosed with the Okubo-
Weiss parameter (Okubo 1970):
$$OW = S_n^2 + S_s^2 - \zeta^2. \tag{4}$$

In this study, the dependence of the energy transfer between the mesoscale eddy and NICs on
the relative vorticity is considered.

**3.2 Modified Slab Model**

A simple linear model known as the slab model (Pollard & Millard 1970) was used in
simulating NIOs in the SML. Analysis of observed currents in the nSCS (to be discussed in Sect.
4) demonstrates that NIOs can also occur in the SML under the nearly steady winds. This suggests
the importance of the energy transfer between the background currents and near-inertial currents
(NICs). To investigate this energy transfer, the background geostrophic currents $(U, V)$ are added
to the original slab model as the modified slab model (Jing et al., 2017; Weller, 1982):
$$\begin{cases} \frac{\partial u}{\partial t} + u\frac{\partial U}{\partial x} + v\frac{\partial U}{\partial y} = fv - ru + \frac{\tau_x}{\rho_o\,H_{mix}} \\ \frac{\partial v}{\partial t} + u\frac{\partial V}{\partial x} + v\frac{\partial V}{\partial y} = -fu - rv + \frac{\tau_y}{\rho_o\,H_{mix}} \end{cases}, \tag{5}$$

where $(u,v)$ are zonal and meridional currents averaged vertically in the SML, and $H_{mix}$ is the
MLD. The damping coefficient $r$ is set to 1/8 days[-1], which is used to parameterize the loss of near-
inertial energy. In Eq. (5), $f$ is the inertial frequency, and $\rho_o$ is the seawater density set to be 1024
kg m[-3]. Wind stress components $(\tau_x, \tau_y)$ are calculated using 10 m height ERA5 winds with the
drag coefficient suggested by Oey et al. (2006).

The above modified slab model uses two important assumptions: the Rossby number of the
geostrophic currents is assumed to be far less than 1, and the horizontal scale of winds to be much
larger than that of mesoscale eddies. By ignoring the background geostrophic currents, the above
modified slab model becomes the original slab model, which was used in many previous studies
of examining the inertial response in the SML (D'Asaro, 1985; Paduan et al. 1989; Pollard &
Millard 1970).



The modified slab model in Eq. (5) can be solved numerically using an implicit numerical
scheme in time to obtain NICs in the SML:

$$
\begin{cases}
\frac{u^{n+1}-u^n}{\Delta t} + u^{n+1}U_x^{n+1} + v^{n+1}U_y^{n+1} = fv^{n+1} - ru^{n+1} + \frac{\tau_x^{n+1}}{\rho H_{mix}} \\
\frac{v^{n+1}-v^n}{\Delta t} + u^{n+1}V_x^{n+1} + v^{n+1}V_y^{n+1} = -fu^{n+1} - rv^{n+1} + \frac{\tau_y^{n+1}}{\rho H_{mix}}
\end{cases}, \tag{6}
$$

where $\Delta t$ is the time step, which is set to 3600 s in this study, and subscripts $x$ and $y$ in $U$ and $V$
represent partial derivatives. The initial value of the NICs is set to 0. In Eq. (6), variables with
superscripts $n$ and $n+1$ represent their values at time $n\Delta t$ and $(n+1)\Delta t$, respectively.
After merging some terms, the numerical scheme of the modified slab model can be written
as

$$
\begin{cases}
\frac{u^{n+1}-u^n}{\Delta t} + a_1^{n+1}u^{n+1} + b_1^{n+1}v^{n+1} = c_1^{n+1} \\
\frac{v^{n+1}-v^n}{\Delta t} + a_2^{n+1}u^{n+1} + b_2^{n+1}v^{n+1} = c_2^{n+1}
\end{cases}, \tag{7}
$$

where $a_1 = U_x + r$, $a_2 = V_x + f$, $b_1 = U_y - f$, $b_2 = V_y + r$, $c_1 = \tau_x/\rho H_{mix}$, $c_2 = \tau_y/\rho H_{mix}$.
Equation (7) can be written in the following tensor form:

$$
\begin{bmatrix} a_1^{n+1} + \frac{1}{\Delta t} & b_1^{n+1} \\ a_2^{n+1} & b_2^{n+1} + \frac{1}{\Delta t} \end{bmatrix} \begin{Bmatrix} u^{n+1} \\ v^{n+1} \end{Bmatrix} = \begin{Bmatrix} \frac{u^n}{\Delta t} + c_1^{n+1} \\ \frac{v^n}{\Delta t} + c_2^{n+1} \end{Bmatrix}. \tag{8}
$$

The numerical update equations for currents are given as

$$
u^{n+1} = \frac{1+\Delta t b_2^{n+1}}{(1+\Delta t a_1^{n+1})(1+\Delta t b_2^{n+1}) - \Delta t b_1^{n+1} a_2^{n+1}} \left[ \frac{u^n}{\Delta t} + c_1^{n+1} - \frac{\Delta t b_1^{n+1}}{1+\Delta t b_2^{n+1}} \left( \frac{v^n}{\Delta t} + c_2^{n+1} \right) \right], \tag{9}
$$

$$
v^{n+1} = \frac{1+\Delta t a_1^{n+1}}{(1+\Delta t a_1^{n+1})(1+\Delta t b_2^{n+1}) - \Delta t b_1^{n+1} a_2^{n+1}} \left[ \frac{u^n}{\Delta t} + c_1^{n+1} - \frac{\Delta t a_2^{n+1}}{1+\Delta t a_1^{n+1}} \left( \frac{v^n}{\Delta t} + c_2^{n+1} \right) \right]. \tag{10}
$$

**3.3 Analysis of NICs**
The observed zonal and meridional currents at 58.3 m below the sea surface at location S2
and at 37.7 m at location S3 (marked in Fig. 1) in the nSCS are analyzed to estimate the near-
inertial kinetic energy in the SML. The observed NICs are obtained by using a band-pass filter
with a frequency band of $0.85f$-$1.15f$.



The surface geostrophic currents described in Sect. 2 are specified in the modified slab model,
based on the assumption that the geostrophic currents are vertically uniform in the SML during
the study period. The simulated currents by the original and modified slab models are band-passed
with the frequency band of $0.85f$-$1.15f$ and are further smoothed using a running window of two
inertial periods to obtain the simulated NICs.
To quantitatively assess performances of the original and modified slab models, the
correlation analysis and root mean square error (RMSE) analysis between the original slab model,
the modified slab model and observations respectively are made.
**4 Results**
**4.1 Observed NICs**
Time series of observed NICs at the top depth bins (58.3 m at S2 and 37.7 m at S3) of two
subsurface ADCP moorings (known as $\vec{u}_{S2}^{top}$ and $\vec{u}_{S3}^{top}$) are shown in black lines in Fig. 2 during
the observational period from day 234 (22 August 2016) to day 492 (7 May 2017) with respect to
1 January 2016. Intense NICs were generated and lasted for about 11 days from day 293 to day
under the largest wind forcing (~1.6 N m$^{-2}$) on day 293 (Fig. 2e and 2f). The largest speed of
NICs was ~0.30 m s$^{-1}$ on day 295 at the top depth bin at mooring S2 (Fig. 2a) and ~0.37 m s$^{-1}$ on
day 296 at the top depth bin at mooring S3 (Fig. 2b). Some NICs were also excited on other days
when the winds were relatively weak and nearly steady.



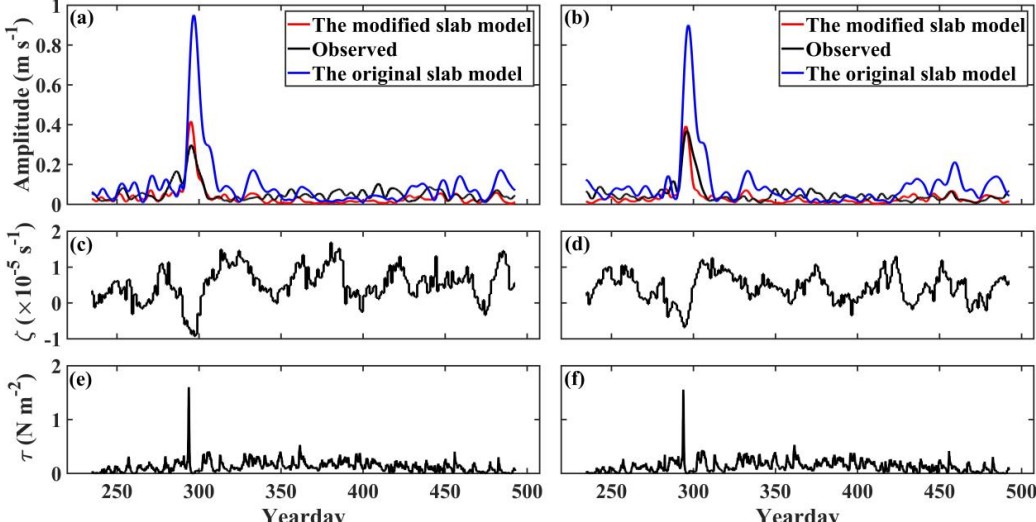

**Figure 2.** Speeds (m s$^{-1}$) of observed (black line) and simulated NICs at the top depth bins of ADCP observations at moorings (a) S2 and (b) S3. Red and blue lines denote model results produced by the modified slab and original slab models respectively. Time series of relative vorticity of the surface geostrophic currents at moorings (c) S2 and (d) S3, and time series of wind stress at moorings (e) S2 and (f) S3. Yearday is the day relative to 00:00:00 (GMT) on 1 January 2016.

Observed NICs at other depth bins of these two ADCP moorings reveal that the intense NICs occurred on days 293-304 at depths between 60 m and 200 m at mooring S2 (not shown) and these NICs were originated from the SML. At mooring S3, similar intense NICs occurred at depths between 40 m and 180 m on days 280-305 (not shown), and these NIC were also originated from the SML. On days 306-318, in comparison, moderate NICs occurred in the lower layer between 150 m and 310 m at these two moorings.

It should be noted that the current observations ($\vec{u}_{S2}^{top}$ and $\vec{u}_{S3}^{top}$) at the top bins of the two ADCP moorings are in the lower part of the SML or below the SML during the observational period. Based on WOA2018, the climatological monthly mean MLD at the two ADCP moorings is the thinnest and ~20 m in August and increases to the maximum value of ~90 m in January. The MLD decreases from ~56 m in February to ~37 m in May and is about 25 m in June and July. This suggests that the observed NICs at the top depth bin (58.3 m) of S2 were made in the lower part of the SML in December and January (days 335-396), but below the SML on the other days of the observational period (Fig. 2a). By comparison, the observed NICs at the top depth bin (37.7 m) of



S3 were made in the middle of the SML in December and January (days 335-396), and in the lower
part of the SML in October and November (days 274-334) and February-May (days 397-492).
The relative vorticity estimated from the surface geostrophic currents was negative at the two
moorings on days 290-300 (Fig. 2c and 2d), with the maximum negative values of about $-0.92\times10^{-5}$ $s^{-1}$
at mooring S2 and about $-0.68\times10^{-5}$ $s^{-1}$ at mooring S3. The negative values of the relative
vorticity during this period resulted from the westward propagation of an anticyclonic eddy. As
shown in Fig. 3a and 3f, on day 295, moorings S2 and S3 were located over the area between a
relatively strong cyclonic eddy to the southwest and two separate weak cyclonic eddies to the east
and south. The anticyclonic eddy moved westward and passed mooring S2 before day 316 (Fig.
3b and 3g). On day 316, the relative vorticity was low and positive of about $1.23\times10^{-5}$ $s^{-1}$ at
mooring S2 and $0.83\times10^{-5}$ $s^{-1}$ at mooring S3 respectively. On days 350-450, an anticyclonic eddy
moved westward and passed through these two moorings S2 and S3, and there was a week cyclonic
eddy close to the two moorings on day 435 (Fig. 3c, 3d, 3h, and 3i). The relative vorticity was
relatively strong and positive during this period, with the maximum positive values of about
$1.68\times10^{-5}$ $s^{-1}$ on day 380 at mooring S2 and $1.29\times10^{-5}$ $s^{-1}$ on day 423 at mooring S3 respectively.
Between days 451-492, there is a week anticyclonic eddy close to the moorings S2 and S3 and the
two moorings are located at the edge of the anticyclonic eddy on day 452 (Fig. 3e and 3j).

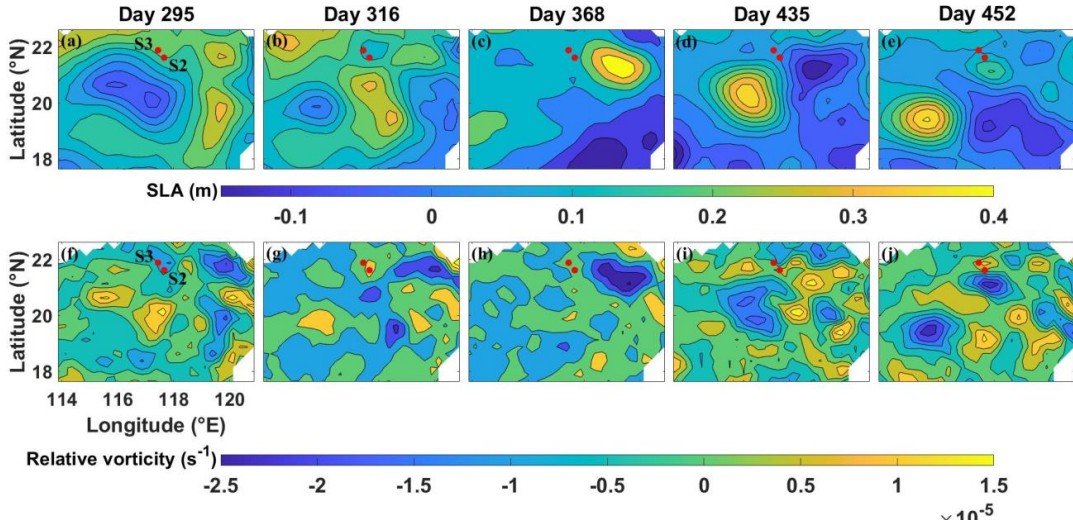


**Figure 3.** Spatial distributions of sea surface level anomaly (SLA) on day 295 (a), day 316 (b),
day 368 (c), day 435 (d) and day 452 (e). Spatial distributions of the relative vorticity on day 295



(f), day 316 (g), day 368 (h), day 435 (i) and day 452 (j). Two red dots mark locations of two
ADCP moorings S2 and S3.

**4.2 Simulated NICs**

Numerical simulations are made using the original and modified slab models to examine
whether the energy exchange occurs between the mesoscale eddy and NICs in the SML at
moorings S2 and S3. Both the original and modified slab models assume that the NICs in the SML
is vertically uniform and use the damping coefficient $r$ set to a relatively small value of 1/8 day$^{-1}$.
Both the models are forced by time series of wind stress shown in Fig. 2e and 2f.
The simulated NICs produced by the original slab model at the top bins of two ADCP
moorings are shown by the blue lines in Fig. 2a and 2b. The simulated NICs by the original slab
model are large and about 0.95 m s$^{-1}$ (0.90 m s$^{-1}$) at mooring S2 (S3) on day 296 and relatively
weak on days 350-430. In comparison with the observed NICs, the original slab model has large
deficiency of overpredicting significantly the observed large NICs on days 285-300 at the two
stations and also overpredicting moderately the observed NICs on other days of the observational
period. It should be noted that the results of both the original and modified slab models using three
different damping coefficients ( $r$ = 1/5 day$^{-1}$, 1/6 day$^{-1}$ and 1/7 day$^{-1}$) and annual mean MLD at
two stations (~45 m) are highly similar with the model results using $r$ = 1/8 day$^{-1}$ and monthly
mean MLD shown in Fig. 2a and 2b.
In comparison with the original slab model results, the modified slab model generates much
smaller NICs than the original slab model on days 280-305 (red lines in Fig. 2a and 2b), with the
maximum value of about 0.42 m s$^{-1}$ at mooring S2 and about 0.39 m s$^{-1}$ at mooring S3 on day 295.
In comparison with the observed NICs at the two moorings, the modified slab model performs
significantly better than the original slab model on days 250-325 and days 451-492, indicating the
importance of the energy exchange between the background mesoscale eddies and NICs. On days
350-450, the relative vorticity of the background currents is positive, which results in the simulated
NICs produced by the modified slab model to be slightly weaker than the observed NICs at these
two moorings. As mentioned in Sect. 4.1, the top depth bins of the ADCP observations at the two
moorings were in the lower part of the SML or below the SML during the observational period,
which explains partially the differences between observed and simulated NICs by the modified
slab model shown in Fig. 2a and 2b. Differences between the observed and simulated NICs can



also partially be explained by the assumption of vertically uniform geostrophic currents in the
SML and exclusion of baroclinic dynamics.
The above analysis based on results shown in Fig. 2 suggests that, overall, the simulated NICs
produced by the modified slab model agree with the observed NICs significantly better than the
original slab model at two moorings S2 and S3. This indicates the occurrence of the near-inertial
energy transfer induced by the interaction between mesoscale eddies and NICs in the SML during
the observational period.
To quantify the model performance, we use the correlation coefficient (R) and the root mean
square error (RMSE) based on the time series of observed and simulated NICs shown in Fig. 2a
and 2b. The modified slab model has a higher correlation coefficient (R = ~0.81) and smaller root
mean square error (RMSE = ~0.04 m s$^{-1}$) than the original slab model (R = ~0.70 and RMSE =
~0.12 m s$^{-1}$) at mooring S2. At mooring S3, the R value between the observed NICs and results of
the modified slab model is ~0.84, and RMSE is ~0.03 m s$^{-1}$. For the original slab model at location
S3, the R value is ~0.85, and RMSE is ~0.10 m s$^{-1}$. These statistical indices suggest that the
modified slab model performs better than the original slab model, especially in reproducing the
amplitude of the observed NICs. This also suggests the energy transfer between mesoscale eddies
and NICs may be a non-negligible process in the energy cascade across different scales in the
global ocean.

**Table 1.** Correlation coefficients (*R*) and RMSEs between the simulation results of the modified
and original slab models and observations for the observational period, respectively.

| Mooring | Correlation coefficient (*R*) | | RMSE (m s$^{-1}$) | |
|---|---|---|---|---|
| | Original slab model | Modified slab model | Original slab model | Modified slab model |
| S2 | 0.70 | 0.81 | 0.12 | 0.04 |
| S3 | 0.85 | 0.84 | 0.10 | 0.03 |

## 5 Sensitivity Study

A series of numerical experiments (in total 196) are conducted using the original and modified
slab models to examine sensitivity of model results to the wind speed, the wind rotation frequency,
the translational speed of the mesoscale eddy, and the strength of the mesoscale eddy. Both the
cyclonic and anticyclonic eddies with an idealized eddy structure are used for simplicity. Results



of the original and modified slab models are both band-pass $(0.60f\text{-}1.40f)$ filtered to get broad
NICs signals and then smoothed using a running window to obtain the near-inertial velocity in the
SML.

**5.1 Idealized Mesoscale Eddy Structure**

Based on the composite analysis of satellite altimetry and Argo float data, Zhang et al. (2013)
suggested a universal structure of mesoscale eddies in the global ocean. Their universal structure
of mesoscale eddies is used in our experiments.
The normalized structure $\tilde{P}(\tilde{r}, z)$ of the pressure anomaly in the universal mesoscale eddy
used in this study is decomposed into a radial function $R(\tilde{r})$ and a vertical function $H(z)$:

$$\tilde{P}(\tilde{r}, z) = R(\tilde{r})\, H(z), \tag{11}$$

$$R(\tilde{r}) = \left(1 - \frac{\tilde{r}^2}{2}\right) e^{-\frac{\tilde{r}^2}{2}}, \tag{12}$$

$$H(z_s) = H_0 \sin(kz_s + \theta_0) + H_{ave}, \tag{13}$$

$$\tilde{r} = \frac{r}{R_0}, \tag{14}$$

$$z_s = \int_0^z \left(\frac{N}{f}\right) dz, \tag{15}$$

where $r$ is the radial distance to the eddy center, $R_0$ is the radius of the mesoscale eddy, N is the
buoyancy frequency, $H_0$, $k$, $\theta_0$ and $H_{ave}$ are undetermined coefficients, e.g. in this study, $H_0 =$
2/3, $H_{ave} = 2/3$, $N = 10^{-3}\ (1/s)$, $k = \pi/18000$, $f = 5 \times 10^{-5}\ (rad/s)$ and $\theta_0 = \pi/6$.
The vertical structure function $H(z)$ defined above is similar with the structural diagram in Zhang
et al. (2013).
The structure function for the idealized mesoscale eddy in the Cartesian coordinate system
can be written as

$$\tilde{P}(x, y, z) = \left(1 - \frac{x^2+y^2}{2R_0^2}\right) \cdot e^{-\frac{x^2+y^2}{2R_0^2}} \cdot \left[\frac{2}{3} \cdot \sin\left(\frac{\pi}{900} \cdot z + \frac{\pi}{6}\right) + \frac{2}{3}\right], \tag{16}$$

with the origin of the coordinate to be the eddy center at the sea surface.
Using the pressure anomaly suggested by Wei et al. (2017), the equation for the pressure field
based on the different strength of mesoscale eddies can be given as

$$P(x, y, z) = P_0 \cdot \tilde{P}(x, y, z) + \bar{P}(z), \tag{17}$$

$$P_0 = g \cdot \rho_0 \cdot SLA_c, \tag{18}$$





where $P_0$ is the strength of mesoscale eddies,$\rho_0$ is the reference density of seawater taken as 1024 kg/m³ , $g$ is the gravitational acceleration set to be 9.8 m/s², $SLA_c$ is the sea surface level anomaly (SLA) of the eddy center where the anticyclone eddies are specified as negative values and the cyclonic eddies are positive values, and the average pressure field is denoted by $\bar{P}$. As only mesoscale eddy signals are added to the ocean in this study, $\bar{P}$ is a function of the vertical direction that is homogeneous in the horizontal direction.

Using the geostrophic balance, the zonal and meridional components of the geostrophic velocity at the ocean surface are given as

$$u = -\frac{1}{\rho_0 f}\frac{\partial P}{\partial y} = -\frac{P_0}{\rho_0 f}\left[\frac{y^3 + yx^2 - 4yR_0{}^2}{2R_0{}^4}\right] e^{-\frac{x^2+y^2}{2R_0{}^2}}, \tag{19}$$

$$v = \frac{1}{\rho_0 f}\frac{\partial P}{\partial x} = \frac{P_0}{\rho_0 f}\left[\frac{x^3 + xy^2 - 4xR_0{}^2}{2R_0{}^4}\right] e^{-\frac{x^2+y^2}{2R_0{}^2}}. \tag{20}$$

In this study, the cyclonic and anticyclonic eddies are set to have the same strength with the opposite relative vorticity. Figure 4 shows currents and relative vorticity at the sea surface for an idealized anticyclonic eddy with the radius of 120 km, $R_0$ to be 120 kilometers, and $P_0$ of 6400 kg/m · s² (i.e., $SLA_c$ is equal to 0.64 m). Based on Eq. (18), the mesoscale eddy strength $P_0$ is a positive proportional function of the $SLA_c$ under the constant seawater density and gravitational acceleration. The strength of the mesoscale eddy can be characterized by the absolute values of the $SLA_c$ ($|SLA_c|$).

To examine model results inside the eddy, nine fixed locations in space named P1-P9 along the y-axis (marked in Fig. 4c) are selected. The zonal and meridional components of currents for an anticyclonic eddy increase from the center to the edge of the eddy, and then gradually decrease outside the anticyclonic eddy (Fig. 4a and 4b). The relative vorticity is largest at the eddy center and then reduces gradually from the center to the edge of the eddy (Fig. 4c). The idealized eddy has a velocity reversal from the eddy center outward, which forms a circular positive (negative) vorticity around the periphery of the idealized anticyclonic (cyclonic) eddy in the Northern Hemisphere (Zhang et.al 2013).



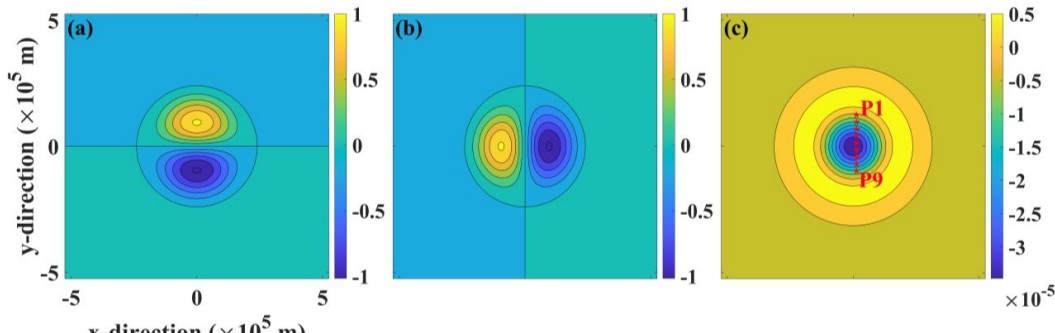

**Figure 4.** Distributions of (a) zonal ($u$) and (b) meridional ($v$) components (m s$^{-1}$) of currents and
(c) relative vorticity (s$^{-1}$) for an idealized anticyclonic eddy with of radius of 120 km. Model results
at nine fixed locations (P1-P9) denoted by red asterisks along the y-axis in (c) are examined.

Jing et al. (2017) proposed a method to calculate the efficiency of energy transfer from
background mesoscale eddies to wind-induced NICs. In this study, we use the differences in the
average speeds of NICs between the modified and original model (NICs_U$_{AE}$ and NICs_U$_{CE}$) as
the proxies for the near-inertial energy generated in the mesoscale eddies by the interaction
between mesoscale eddies and NICs:

$$NICs\_U_{AE} = NICs\_U_{AE}^{Modified} - NICs\_U_{AE}^{Original}, \tag{21}$$

$$NICs\_U_{CE} = NICs\_U_{CE}^{Modified} - NICs\_U_{CE}^{Original}, \tag{22}$$

where $NICs\_U_{AE}^{Modified}$ ($NICs\_U_{AE}^{Original}$) and $NICs\_U_{CE}^{Modified}$ ($NICs\_U_{CE}^{Original}$) are the averaged
speeds of NICs based on results produced by the modified (original) slab model in the anticyclonic
eddies and cyclonic eddies over the same time period, respectively. It should be noted that the
simulated NICs by the original slab model represent the near-inertial energy generated directly by
the wind forcing. Therefore, after removing the generation of wind-induced NICs, the differences
$NICs\_U_{AE}$ and $NICs\_U_{CE}$ represent the amplitudes of NICs transferred by interactions between
mesoscale eddies and NICs in the anticyclonic eddies and the cyclonic eddies respectively.

To quantify differences in the NICs transferred by background currents between the
anticyclonic and cyclonic eddies, we introduce a simple parameter $\alpha$, defined as

$$\alpha = \frac{NICs\_U_{AE}}{NICs\_U_{CE}}. \tag{23}$$




If $\alpha$ is larger than 1, it means that the anticyclonic eddy can transfer more NICs than the cyclonic
eddy with the same strength.

**5.2 Effect of Wind Speeds**

The wind speed affects the energy input from the wind to the SML, and therefore influences
interaction between mesoscale eddies and NICs. To facilitate theoretical analysis and generate a
reasonable magnitude of the NICs speeds, we conduct the numerical experiments using
cyclonically rotating winds and anticyclonically rotating and in the Northern Hemisphere with the
constant wind speed (A). The wind stress $\tau$ used in our sensitivity study takes a form as follows
$$\tau_x(t) + i\,\tau_y(t) = A\,e^{iBt}, \tag{24}$$

where $\tau_x(t)$ and $\tau_y(t)$ are time-dependent zonal and meridional components of wind stress, and
$B$ is the wind rotation frequency. Positive wind rotation frequencies represent cyclonically
rotating winds and negative wind rotation frequencies indicate anticyclonically rotating winds.
Five numerical experiments (ExpA1-5) are conducted with the background idealized
mesoscale eddy moving westward with a translational speed of 8 cm s$^{-1}$. The $|SLA_c|$ of both
anticyclonic and cyclonic eddies are set to 0.64 m in these five experiments. The speeds (A) of
time-varying winds in these five experiments are set to 5 m s$^{-1}$, 10 m s$^{-1}$, 13 m s$^{-1}$, 15 m s$^{-1}$, and 20
m s$^{-1}$ respectively, corresponding to wind stress amplitudes of 0.038 N m$^{-2}$, 0.150 N m$^{-2}$, 0.282 N
m$^{-2}$, 0.412 N m$^{-2}$, and 0.876 N m$^{-2}$ respectively. The wind forcing rotates cyclonically at the inertial
frequency $f$.
For a cyclonic eddy with $|SLA_c| = 0.64$ m, the sum of averaged speeds of NICs converted
from this cyclonic eddy at the above-mentioned nine locations ($\sum \text{NICs\_U}_{CE}$) are about 0.009 m s$^{-1}$
$^{1}$ and 0.209 m s$^{-1}$ for cyclonic wind speeds of 5 m s$^{-1}$ (i.e. 0.038 N m$^{-2}$) and 20 m s$^{-1}$ (i.e. 0.876 N
m$^{-2}$) respectively (Fig. 5a). This suggests that, within the cyclonic eddy, the $\sum \text{NICs\_U}_{CE}$ increases
23 times if the cyclonic wind stress increases 23 times, which is consistent with the conclusion
based on the analytical solution in Sect. 6.
In an anticyclonic eddy with the same strength of $|SLA_c| = 0.64$ m, the sum of averaged
speeds of NICs transferred from this anticyclonic eddy at the nine locations ($\sum \text{NICs\_U}_{AE}$) also
increases with the wind speeds. The $\sum \text{NICs\_U}_{AE}$ values are about 0.019 m s$^{-1}$ and 0.437 m s$^{-1}$ for
the cyclonic wind speeds of 5 m s$^{-1}$ (corresponding to 0.038 N m$^{-2}$) and 20 m s$^{-1}$ (corresponding to





0.876 N m$^{-2}$) respectively (Fig. 5a). This indicates that the averaged speeds of NICs generated in
anticyclonic eddies by the interaction between background anticyclonic eddies and NICs also
increase 23 times if the cyclonic wind stress increases 23 times. But the NICs are stronger in the
anticyclonic eddy than in the cyclonic eddy under the same wind conditions.

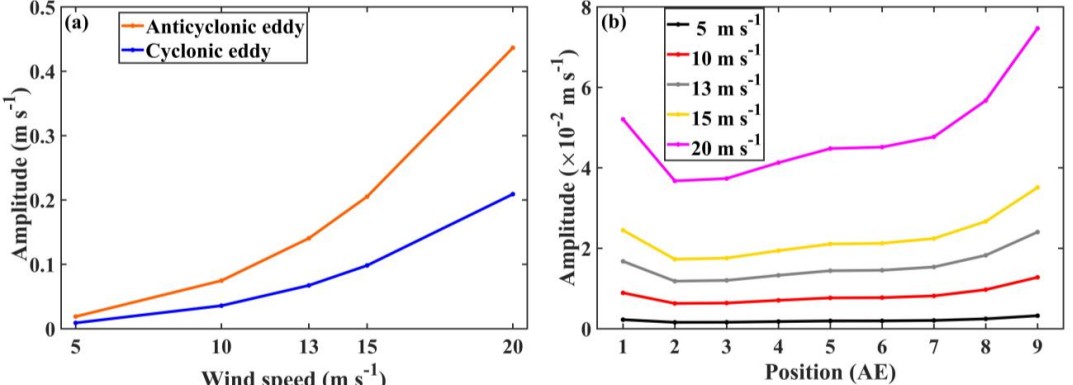


**Figure 5.** (a) Sum of averaged speeds of transferred NICs at 9 fixed locations P1-P9 as a function
of the wind speeds in the anticyclonic eddy (orange line) and cyclonic eddy (blue line), respectively.
(b) Averaged speeds of transferred NICs as a function of the wind speeds in the anticyclonic eddy.
The black, red, gray, yellow and purple line indicate respectively the wind speed of 5 m s$^{-1}$ (ExpA1),
10 m s$^{-1}$ (ExpA2), 13 m s$^{-1}$ (ExpA3), 15 m s$^{-1}$ (ExpA4) and 20 m s$^{-1}$ (ExpA5). Numbers on the
horizontal axis in (b) denote nine fixed locations P1 to P9. The wind rotates cyclonically at the
inertial frequency. Mesoscale eddies move westward at the translational speed of 8 cm s$^{-1}$ and
$|SLA_c| = 0.64$ m.


It should be noted that, however, the transferred near-inertial energy varies with the actual

locations within the mesoscale eddy. We take the example of anticyclonic eddies to illustrate this
issue. In anticyclonic eddies, the difference between nine locations P1-P9 is large (Fig. 5b). The
amplitude shows a distribution characterized by small values at the eddy center and relatively large
values at the eddy edge, and this distribution characteristic is more obvious with the increase of
the wind speed. When the wind speed is relatively small, the difference of the energy generation
induced by the Okubo-Weiss parameter is not significant. With the increase of the wind energy
input, the larger absolute value of the positive Okubo-Weiss parameter gradually has the decisive
function in the energy transfer. Therefore, it makes the anticyclonic eddy exhibit superior energy





conversion characteristics at the eddy edge, which is consistent with the conclusion based on the
energy transfer rate (Fig. 10).

The $\alpha$ values (Eq. (23)) are about 2.08 based on the sum of averaged speeds of NICs at the

nine locations in the five different wind speeds. This indicates that the anticyclonic eddy is more
efficient than the cyclonic eddy in transferring the kinetic energy to NICs (Fig. 5a). The difference
in the near-inertial energy transfer efficiency between anticyclonic and cyclonic eddies is not
affected very much by magnitudes of wind speeds.

**5.3 Effect of Wind Rotation Frequencies**

The rotation frequency of the winds can affect the generation of the NICs and thus the energy

transfer between the mesoscale eddy and NICs, therefore six numerical experiments using different
wind rotation frequencies (-1.5$f$, -1.25$f$, -1.1$f$, -0.9$f$, -0.75$f$, -0.5$f$, -0.25$f$, 0.25$f$, 0.5$f$, 0.75$f$,
$f$, 1.25$f$ and 1.5$f$, where $f$ is the inertial frequency), denoted as ExpB1-13, are conducted.
Positive wind rotation frequencies correspond to cyclonically rotating winds, and negative wind
rotation frequencies are for anticyclonically rotating winds. In these thirteen experiments, the
mesoscale eddy moves westward at the speed of 8 cm s$^{-1}$ and $|SLA_c|$ = 0.64 m. The winds rotate
at different frequencies and the wind speed is set to 13 m s$^{-1}$.

Figure 6 shows the sum of averaged speeds of the transferred NICs at the nine locations as

function of the wind rotation frequency for the cyclonic and anticyclonic eddies. Under
cyclonically rotating and anticyclonically rotating wind conditions, there is bidirectional energy
transfer between mesoscale eddies and NICs. The closer the absolute value of the wind rotation
frequency to $f$, the stronger the energy transfer between mesoscale eddies and NICs.

For cyclonic eddies, the sum of averaged speeds of NICs ($\sum \text{NICs\_U}_{CE}$) is sensitive to the

wind rotation frequency. The $\sum \text{NICs\_U}_{CE}$ values are less than zero when the winds rotate
cyclonically at the frequencies of 0.25$f$, 0.75$f$ and 1.25$f$ (Fig. 6b) and the winds rotate
anticyclonically at the frequencies of -1.1$f$, -0.9$f$ and -0.75$f$ (Fig. 6a), indicating that direction of
the energy transfer is from the NICs to the cyclonic eddies. The amplitudes of the energy
transferred from NICs to cyclonic eddies under anticyclonically rotating wind conditions are larger
than those transferred under cyclonically rotating wind conditions. The addition of mesoscale
eddies has a damping effect for NICs, leading to the negative energy transfer that aligns with the
observed results (Fig. 2). When the direction of the energy transfer is positive, the $\sum \text{NICs\_U}_{CE}$ has




a maximum value of about 0.635 m s$^{-1}$ when the winds rotate anticyclonically at the frequency of
-1.25$f$.

For anticyclonic eddies, the sum of averaged speeds of NICs ($\sum$ NICs_U$_{AE}$) also varies with

the wind rotating frequency. When the winds rotate anticyclonically at frequencies of -1.5$f$, -1.25$f$,
-1.1$f$ and -0.9$f$, the direction of the energy transfer is from NICs to anticyclonic eddies (Fig. 6a).
The negative energy transfer is strongest at the frequency of -1.1$f$. The $\sum$ NICs_U$_{AE}$ values are all
positive under cyclonically rotating winds, which represent the energy transfer from the
anticyclonic eddies to NICs (Fig. 6b). The $\sum$ NICs_U$_{AE}$ values for the anticyclonic eddies increases
from the value of ~0.002 m s$^{-1}$ at the wind rotating frequency of 0.25$f$ to the maximum value of
~0.162 m s$^{-1}$ at the wind rotating frequency of 0.75$f$. The $\sum$ NICs_U$_{AE}$ values are larger than the
$\sum$ NICs_U$_{CE}$ values under the same positive wind rotation frequency. The closer the rotational
frequency of the cyclonic winds is to the inertial frequency, the greater the difference in near-
inertial energy conversion induced by anticyclonic eddies and cyclonic eddies.

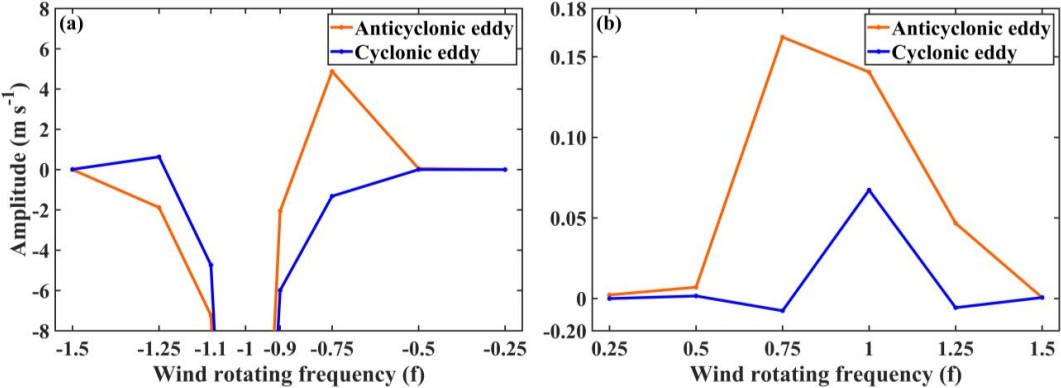


**Figure 6.** Sum of averaged speeds of transferred NICs at 9 fixed locations P1-P9 as a function of
rotation frequencies of (a) anticyclonically rotating winds and (b) cyclonically rotating winds. The
orange and blue line indicate respectively the anticyclonic eddy and cyclonic eddy. The wind
rotation frequencies are normalized by the inertial frequency $f$.

**5.4 Effect of Eddy Translational Speeds**

The translational speed of a background mesoscale eddy defines the forcing duration of winds

and thus the energy input to NICs in the ocean SML. Based on the observations of mesoscale





459 eddies in the nSCS, nine numerical experiments (ExpC1-9) using different translational speeds of

460 mesoscale eddies (4 cm s$^{-1}$, 5 cm s$^{-1}$, 6 cm s$^{-1}$, 7 cm s$^{-1}$, 8 cm s$^{-1}$, 9 cm s$^{-1}$, 10 cm s$^{-1}$, 11 cm s$^{-1}$, and

461 12 cm s$^{-1}$) are conducted. In these nine experiments, the speed of cyclonically rotating winds at the

462 inertial frequency is set to 13 m s$^{-1}$. The mesoscale eddy moves westward and $|SLA_c| = 0.64$ m.

463 For the anticyclonic eddy with $|SLA_c| = 0.64$ m, the sum of average speeds of the converted

464 NICs at the nine locations ($\sum$ NICs_U$_{AE}$) increases from about 0.133 m s$^{-1}$ to 0.148 m s$^{-1}$ as the

465 translational speeds increase from 4 cm s$^{-1}$ to 12 cm s$^{-1}$ (Fig. 7a). The increase of the translational

466 speed enhances the total kinetic energy of mesoscale eddies, which can provide a larger energy

467 source and be more beneficial for the conversion of NICs. It should be noted that the change in the

468 eddy kinetic energy caused by the different translational speeds is relatively small in comparison

469 with the total eddy energy determined by the mesoscale eddy strength. Therefore, the change in

470 the amplitude of the transferred NICs is relatively small.

471 For a cyclonic eddy with $|SLA_c| = 0.64$ m, the sum of average speeds of the transferred NICs

472 at the nine locations ($\sum$ NICs_U$_{CE}$) range from 0.066 m s$^{-1}$ to 0.069 m s$^{-1}$ and are not sensitive to

473 the translational speeds (Fig. 7a). Same as in the anticyclonic eddy case, more total kinetic energy

474 is available for generating NICs within the cyclonic eddy with the same structure but with the

475 faster translational speeds. Different from the anticyclonic eddy case, however, the anticyclonic

476 eddy is not conducive to the energy transfer to NICs naturally. Therefore, the slightly larger energy

477 source caused by the increase of the translational speeds has little influence on the total amount of

478 kinetic energy transferred from the cyclonic eddy to NICs.

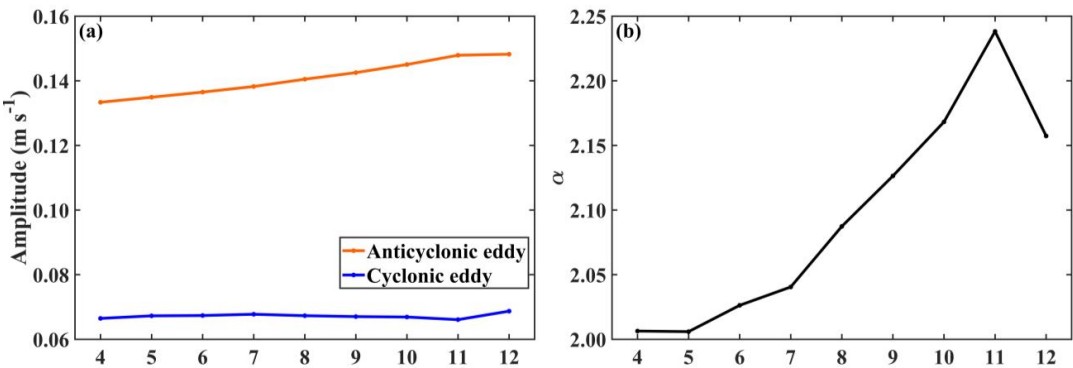



**Figure 7.** (a) Sum of averaged speeds of transferred NICs at 9 fixed locations P1-P9 as a function of the eddy translational speed in the anticyclonic eddy (orange line) and cyclonic eddy (blue line), respectively. (b) The $\alpha$ value as a function of the eddy translational speed. The speed of the cyclonic wind is 13 m s$^{-1}$, and the wind rotates at the inertial frequency. Mesoscale eddies move westward and $|SLA_c|$ = 0.64 m.

The $\alpha$ value has a maximum value of ~2.24 occurring at the eddy translational speeds of 11 cm s$^{-1}$ (Fig. 7b). This means that the anticyclonic eddy transfers much more near-inertial energy than the cyclonic eddy does, particular at the translational speed of 11 cm s$^{-1}$. After exceeding the translational speed of 11 cm s$^{-1}$, the $\alpha$ values decrease with the increase of the eddy translational speeds. The $\alpha$ value is ~2.16 at the translational speed of 12 cm s$^{-1}$.

A natural question raises whether the variations of $\alpha$ values within the mesoscale eddy are affected by the strength of anticyclonic eddies. To address this issue, we consider the sum of averaged speeds of NICs at nine locations for mesoscale eddies and the $\alpha$ values with different strengths and translational speeds (Fig. 8). We consider the case with the cyclonically rotating wind speed of 13 m s$^{-1}$ at the inertial frequency. For mesoscale eddies with larger $|SLA_c|$ values, the $\alpha$ values are relatively less sensitive to the translational speed. For mesoscale eddies with different $|SLA_c|$ values, the $\alpha$ values all have the maximum values with the translational speeds of 11 cm s$^{-1}$. The $\alpha$ value decreases with the elevated translational speed when the eddy translational speed is larger than 11 cm s$^{-1}$.





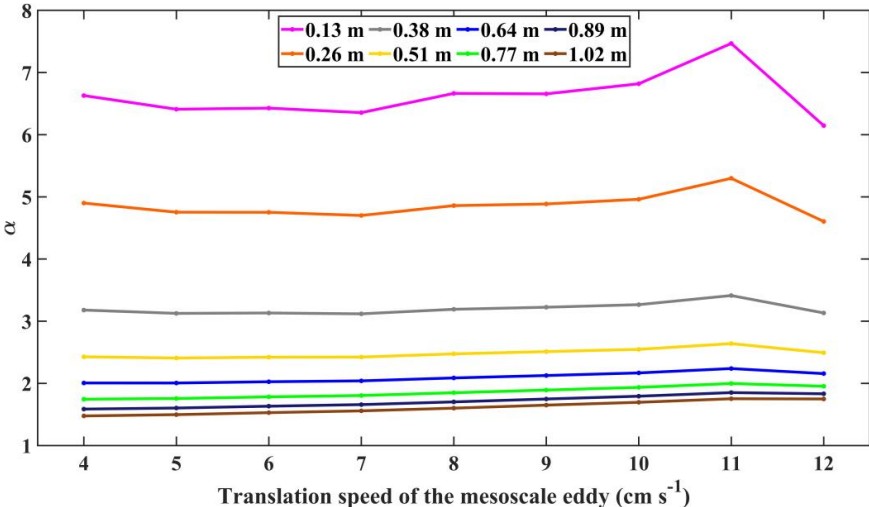

**Figure 8.** The $\alpha$ values at 9 fixed locations P1-P9 as a function of the strengths and translational
speeds of the mesoscale eddies. The speed of the cyclonically rotating winds at the inertial
frequency is 13 m s$^{-1}$. Mesoscale eddies move westward. Different colors of lines represent
different mesoscale eddy strengths.

## 5.5 Effect of Mesoscale Eddy Strengths

In addition to the effect of the eddy translational speed, other characteristics of mesoscale
eddies such as the radius and strength of mesoscale eddies can also affect the energy exchange.
Sixteen numerical experiments (denoted as ExpD1-16) are conducted in this section using various
strengths of mesoscale eddies. In these sixteen experiments, the speed of cyclonically rotating
winds at the inertial frequency is set to 13 m s$^{-1}$, the translational speed of mesoscale eddies is set
to 8 cm s$^{-1}$, and the eddy translational direction to be westward. The $|SLA_c|$ values are set to 0.13
m, 0.26 m, 0.38 m, 0.51 m, 0.64 m, 0.77 m, 0.89 m, and 1.02 m for cyclonic and anticyclonic
eddies, respectively.

Figure 9a shows that the sum of averaged speeds of the transferred NICs at nine locations as
a function of the mesoscale eddy strengths for the cyclonic ($\sum \text{NICs\_U}_{CE}$) and anticyclonic eddies
($\sum \text{NICs\_U}_{CE}$). Both values of $\sum \text{NICs\_U}_{AE}$ and $\sum \text{NICs\_U}_{CE}$ are larger for higher eddy strengths,
particularly for the anticyclonic eddies. For the $|SLA_c|$ values equal to 0.13 m, values of
$\sum \text{NICs\_U}_{AE}$ and $\sum \text{NICs\_U}_{CE}$ are about 0.019 m s$^{-1}$ and about 0.003 m s$^{-1}$. The averaged speeds of



the converted NICs increase with the $|SLA_c|$ value, particularly for the anticyclonic eddies. For $|SLA_c|$ = 1.02 m, the averaged speeds of the transferred NICs are ~0.280 m s$^{-1}$ and ~0.175 m s$^{-1}$ for the anticyclonic and cyclonic eddies respectively, which are ~14.74 times and ~58.33 times larger than the counterparts at $|SLA_c|$= 0.13 m. As the geostrophic strain field is relatively stronger at the eddy edge, the average speeds of the converted NICs at the mesoscale eddy edge are larger than that at the mesoscale eddy center.

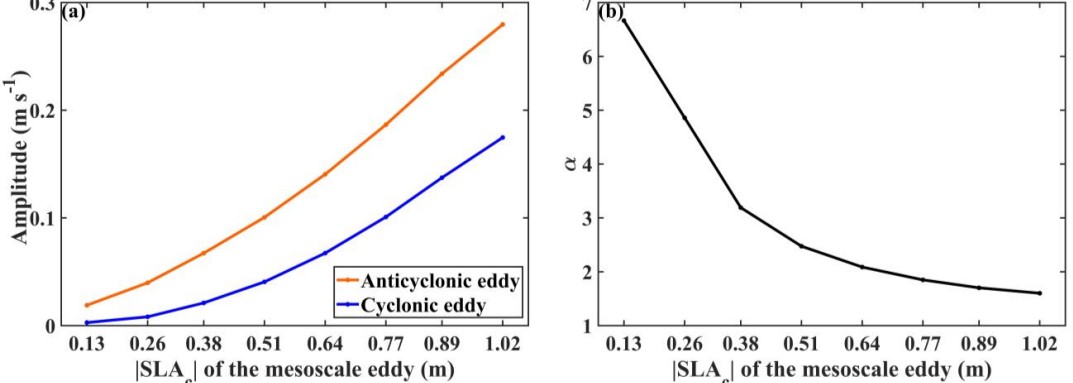

**Figure 9.** (a) Sum of averaged speeds of transferred NICs at 9 fixed locations P1-P9 as a function of the strengths ($|SLA_c|$ of the mesoscale eddy) of the anticyclonic eddy (orange line) and cyclonic eddy (blue line). (b) The $\alpha$ value as a function of the eddy strengths ($|SLA_c|$ of the mesoscale eddy). The mesoscale eddy moves westward at the speed of 8 cm s$^{-1}$. The speed of the cyclonically rotating wind speed at the inertial frequency is set to 13 m s$^{-1}$.

The $\alpha$ value also varies with the mesoscale eddy strength (Fig. 9b). The $\alpha$ value decreases significantly from about 6.66 to 1.60 for the $|SLA_c|$ values in the range of 0.13 and 1.02 m. As mentioned above, cyclonic eddies have limited ability in transferring their kinetic energy to NICs, which differs significantly from anticyclonic eddies. However, stronger cyclonic eddies with more eddy energy provide the more favorable condition for the energy transfer, which can narrow the difference in the near-inertial energy transfer induced by anticyclonic eddies and cyclonic eddies. Furthermore, stronger geostrophic currents lead to stronger geostrophic strain field which can generate stronger NICs.





## 5.6 Relative Vorticity and Strain


As mentioned above, the anticyclonic eddies in the SML are more efficient than cyclonic
eddies for transferring kinetic energy to NICs, which can be explained by the relative vorticity of
the background flow ($\zeta$) defined in Eq. (1). In numerical experiments, the direction of the energy
transfer is bidirectional, but primarily positive, that is, from mesoscale eddies to NICs. For
cyclonic eddies, the direction of the energy transfer is from the NICs to the cyclonic eddy under
cyclonically rotating winds at frequencies of $0.25f$, $0.75f$ and $1.25f$ and anticyclonically rotating
winds at frequencies of $-1.1f$, $-0.9f$ and $-0.75f$. When the frequencies of anticyclonically rotating
winds are range from $-1.5f$ to $-0.9f$, the energy transfer is also negative in the anticyclonic eddy.
The $\alpha$ value is more than 1.0 for about 88% of these experiments, which indicates that the
transferred near-inertial energy is larger in anticyclonic eddies than cyclonic eddies.
In addition to the relative vorticity and translational speed of a mesoscale eddy, the normal
strain and shear strain of the background flow can also affect the energy transfer between the
mesoscale eddy and NICs in the SML. Jing et al. (2017) proposed a method to calculate the rate
of energy transfer from background mesoscale eddies to wind-induced NICs. Following Jing et al.
(2017), the energy transfer rate ($\varepsilon$) between the NICs and the mesoscale eddy in the SML is given
as

$$\varepsilon = -\rho H_{mix}\left(uuU_x + uvU_y + uvU_x + vvV_y\right), \tag{25}$$

where $u$ and $v$ are respectively the zonal and meridional components of the near-inertial current
velocity reproduced by the modified slab model, and subscripts $x$ and $y$ in $U$ and $V$ represent partial
derivatives. The positive $\varepsilon$ mean the energy transfer from the mesoscale eddy to the NICs, and the
negative $\varepsilon$ indicates the backward energy cascade. In the numerical experiments, the NICs can be
generated directly by the cyclonic winds, and the wind-induced NICs can further interact with the
mesoscale eddy and transfer the near-inertial energy from the mesoscale eddy to the NICs when
the mesoscale eddy passed by the nine locations P1-P9. Therefore, we also calculate the energy
transfer rate and Okubo-Weiss parameter at the nine fixed locations P1-P9 in the sensitivity
experiments ExpA3, ExpC1-C9 and ExpD1-D16, which are under the same wind conditions.
When the Okubo-Weiss parameters are negative, the energy transfer rate decreases as the
Okubo-Weiss parameter increases. However, when the Okubo-Weiss parameters are positive, the
energy transfer rate shows an elevated trend with the increase of the Okubo-Weiss parameter (Fig.



10). Based on limited sensitivity studies, we found the relative vorticity and the strain of the
mesoscale eddy both have an influence on the near-inertial energy transferred by interactions
between mesoscale eddies and NICs.

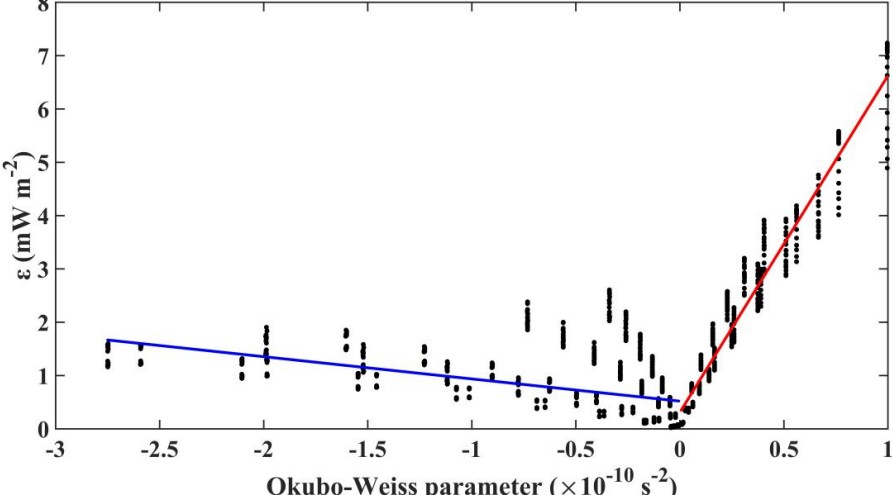


**Figure 10.** Scatterplot between the energy transfer rate and the Okubo-Weiss parameter. The blue
(red) line is the linear fitting line when the Okubo-Weiss parameters are negative (positive).

**6 Theoretical Analyses**
**6.1 Solutions in the Frequency Domain**
To gain better understanding of the role of the relative vorticity in the background flow for
anticyclonic eddies to be significantly more efficient than the cyclonic eddies in transferring their
kinetic energy to NICs, we examine analytically the effect of the relative vorticity in the frequency
domain solution of the modified slab model.
The modified slab model can be written in the tensor form:
$$\frac{\partial}{\partial t}\begin{Bmatrix} u \\ v \end{Bmatrix} + \begin{bmatrix} a_1 & b_1 \\ a_2 & b_2 \end{bmatrix}\begin{Bmatrix} u \\ v \end{Bmatrix} = \begin{Bmatrix} c_1 \\ c_2 \end{Bmatrix}, \tag{26}$$

where, $a_1 = U_x + r$, $a_2 = V_x + f$, $b_1 = U_y - f$, $b_2 = V_y + r$, $c_1 = \tau_x/\rho H_{mix}$ and $c_2 = \tau_y/$
$\rho H_{mix}$. For simplicity, we consider the steady wind forcing to eliminate the wind-induced NICs



here. In the case of steady winds, we use the Fourier transform to translate the modified slab model
from the time domain into the frequency domain:

$$\begin{bmatrix} a_1 + i\omega & b_1 \\ a_2 & b_2 + i\omega \end{bmatrix} \begin{Bmatrix} \tilde{u} \\ \tilde{v} \end{Bmatrix} = \begin{Bmatrix} \tilde{c_1} \\ \tilde{c_2} \end{Bmatrix}, \tag{27}$$

where $\omega$ is the frequency and variables with a tilde represent the values after Fourier transform.

Assuming the mesoscale eddy is in an almost steady state during an inertial period (Jing et al.

2017), the analytical solution for the zonal and meridional components of NICs in the frequency
domain can be written as

$$\tilde{u} = \frac{(b_2+i\omega)\cdot 2\cdot\pi\cdot c_1\cdot\delta(\omega) - b_1\cdot 2\cdot\pi\cdot c_2\cdot\delta(\omega)}{(a_1+i\omega)(b_2+i\omega) - a_2 b_1}, \tag{28}$$

$$\tilde{v} = \frac{(a_1+i\omega)\cdot 2\cdot\pi\cdot c_2\cdot\delta(\omega) - a_2\cdot 2\cdot\pi\cdot c_1\cdot\delta(\omega)}{(a_1+i\omega)(b_2+i\omega) - a_2 b_1}, \tag{29}$$

where $\delta(\omega)$ is the Dirac Delta function.

Based on Perceval's theorem, the energy of NICs is the same in both the time and frequency

domains:

$$\frac{1}{T}\int |U(t)|^2 dt = \int |U(\omega)|^2 d\omega, \tag{30}$$

where $T$ is the total period.

The time-mean near-inertial kinetic energy $\overline{U_{NIW_s}}^2$ in the time domain can be written as

$$\overline{U_{NIW_s}}^2 = \overline{u^2} + \overline{v^2}, \tag{31}$$

$$\overline{u^2} = \int \frac{((b_2 c_1 - b_1 c_2)^2 + \omega^2 c_1{}^2)\cdot((2\cdot\pi\cdot\delta(\omega))^2)}{(a_2 b_1 - a_1 b_2 + \omega^2)^2 + \omega^2(a_1+b_2)^2} \cdot d\omega, \tag{32}$$

$$\overline{v^2} = \int \frac{((a_1 c_2 - a_2 c_1)^2 + \omega^2 c_2{}^2)\cdot((2\cdot\pi\cdot\delta(\omega))^2)}{(a_2 b_1 - a_1 b_2 + \omega^2)^2 + \omega^2(a_1+b_2)^2} \cdot d\omega. \tag{33}$$

For a unidirectional laterally sheared geostrophic flow and the southwestward wind, U=0,

V=V(x), $c_1 < 0$ and $c_2 < 0$. Substitution of $a_1 = r$,$a_2 = \zeta + f$,$b_1 = -f$ and $b_2 = r$ into Eq.
(32) and Eq. (33) yields

$$\overline{u^2} = \int \frac{((rc_1 + fc_2)^2 + \omega^2 c_1{}^2)\left((2\cdot\pi\cdot\delta(\omega))^2\right)}{\left(r^2 - \omega^2 + f(\zeta+f)\right)^2 + 4\omega^2 r^2} d\omega, \tag{34}$$

$$\overline{v^2} = \int \frac{((rc_2 - (\zeta+f)c_1)^2 + \omega^2 c_2{}^2)\left((2\cdot\pi\cdot\delta(\omega))^2\right)}{\left(r^2 - \omega^2 + f(\zeta+f)\right)^2 + 4\omega^2 r^2} d\omega. \tag{35}$$

Since the relative vorticity is negative in anticyclonic eddies, the denominator term for $\overline{u^2}$ in

Eq. (34) is less than the value with the relative vorticity to be positive. Therefore, the value of $\overline{u^2}$
is greater in anticyclonic eddies than in cyclonic eddies. For the positive relative vorticity, the





numerator term of $\overline{v^2}$ is smaller and the denominator term in Eq. (35) becomes larger than the case
of negative vorticity. This indicates that $\overline{v^2}$ is more elevated in the anticyclonic eddies. Since both
$\overline{u^2}$ and $\overline{v^2}$ are elevated when the relative vorticity is negative than counterparts with the positive
relative vorticity, anticyclonic eddies can transfer more near-inertial energy than cyclonic eddies.
**6.2 Analytical Solution**
An analytical solution based on the modified slab model is considered here to demonstrate
that mesoscale eddies can transfer more near-inertial energy for stronger winds. The modified
slab model cane be written as

$$\frac{\partial u}{\partial t} + a_1 u + b_1 v = c_1, \tag{36}$$

$$\frac{\partial v}{\partial t} + a_2 u + b_2 v = c_2, \tag{37}$$

where, $a_1 = U_x + r$, $a_2 = V_x + f$, $b_1 = U_y - f$, $b_2 = V_y + r$, $c_1 = \tau_x / \rho H_{mix}$, $c_2 = \tau_y / \rho H_{mix}$,
$\tau_x = A cos ft$ and $\tau_y = A sin ft$.
For the cyclonic wind, substitution of Eq. (36) into Eq. (37) yields

$$\frac{\partial^2 u}{\partial t^2} + (a_1 + b_2)\frac{\partial u}{\partial t} + (a_1 b_2 - a_2 b_1)u = b_2 c_1 - (f + b_1)c_2 \tag{38}$$

$$\Delta = (a_1 + b_2)^2 - 4(a_1 b_2 - a_2 b_1). \tag{39}$$

The analytical solutions of the current to the modified slab model are

$$u_{Modified} = e^{\gamma t}(Q_1 cos\beta t + Q_2 sin\beta t) + Q_3 cos ft + Q_4 sin ft, \tag{40}$$

$$v_{Modified} = \frac{1}{b_1}(c_1 - e^{\gamma t}(a_1 + \gamma)(Q_1 cos\beta t + Q_2 sin\beta t)$$

$$-e^{\gamma t}\beta(Q_2 cos\beta t - Q_1 sin\beta t) + (fQ_3 - a_1 Q_4)sin ft - (fQ_4 + a_1 Q_3)cos ft), \tag{41}$$

where

$$\gamma = -\frac{a_1 + b_2}{2}, \tag{42}$$

$$\beta = \frac{\sqrt{-\Delta}}{2}, \tag{43}$$

$$Q_1 = -Q_3, \tag{44}$$

$$Q_2 = \frac{c_1 + \gamma Q_3 - fQ_4}{\beta}, \tag{45}$$

$$Q_3 = \frac{Ab_2(a_1 b_2 - a_2 b_1 - f^2) + A(f + b_1)(fa_1 + fb_2)}{\rho H_{mix}((a_1 b_2 - a_2 b_1 - f^2)^2 + (fa_1 + fb_2)^2)}, \tag{46}$$

$$Q_4 = \frac{\rho H_{mix}Q_3(fa_1 + fb_2) - A(f + b_1)}{\rho H_{mix}(a_1 b_2 - a_2 b_1 - f^2)}. \tag{47}$$

 

Increasing the wind stress $c_1$ and $c_2$ by a factor of $n$ named $c_1'$ and $c_2'$ yields
$$c_1' = nc_1 = \frac{n\tau_x}{\rho H_{mix}} = \frac{nA\cos ft}{\rho H_{mix}}, \tag{48}$$

$$c_2' = nc_2 = \frac{n\tau_y}{\rho H_{mix}} = \frac{nA\sin ft}{\rho H_{mix}}. \tag{49}$$

Substitution of $c_1'$ and $c_2'$ into $Q_3$ and $Q_4$ yields
$$Q_3' = \frac{nAb_2(a_1b_2-a_2b_1-f^2)+nA(f+b_1)(fa_1+fb_2)}{\rho H_{mix}((a_1b_2-a_2b_1-f^2)^2+(fa_1+fb_2)^2)} = nQ_3, \tag{50}$$

$$Q_4' = \frac{\rho H_{mix} nQ_3(fa_1+fb_2)-nA(f+b_1)}{\rho H_{mix}(a_1b_2-a_2b_1-f^2)} = nQ_4. \tag{51}$$

Substitution of $Q_3$ and $Q_4$ into $Q_1$ and $Q_2$ yields
$$Q_1' = -nQ_3 = nQ_1, \tag{52}$$

$$Q_2' = \frac{nc_1+\gamma nQ_3-fnQ_4}{\beta} = nQ_2. \tag{53}$$

Therefore, the current with the increased wind stress to the modified slab model is given as
$$u_{Modified}' = nu_{Modified}, \tag{54}$$

$$v_{Modified}' = nv_{Modified}. \tag{55}$$

The analytical solution of the current to the original slab model is
$$U_{Original} = \frac{Ae^{ift}}{\rho H_{mix}(if+r)} + \left(\left(\frac{Ae^{-(if+r)t}}{\rho H_{mix}(if+r)}\right)\left(\frac{if}{2if+r}\right)\left(1 - e^{(2if+r)t}\right)\right) - \frac{Ae^{-(if+r)t}}{\rho H_{mix}(if+r)}. \tag{56}$$

Therefore, the current with the increased wind stress to the original slab model is given as
$$U_{Original}' = nU_{Original}. \tag{57}$$

The above analytical solutions demonstrate that when the wind stress increases by $n$ times,
the current speeds simulated by the modified and original slab models both increase by $n$ times.
The differences in the average speeds of NICs between the modified and original model represent
the transferred near-inertial energy by the interaction between mesoscale eddies and near-inertial
motions (Eq. (21) and (22)). As the NICs are the component of the total currents in the near-inertial
frequency band, the transferred near-inertial energy in the mesoscale eddies also increases by a
factor of n times when the wind stress is multiplied by n times. This feature is consistent with our
sensitivity experiments in Sect. 5.
**7 Summary and Discussion**
Analysis of in situ current observations at two offshore ADCP mooring sites in the northern
South China Sea (nSCS) demonstrated that relatively strong near-inertial currents (NICs) occurred



during certain periods of nearly steady winds in the lower part of the ocean surface mixed layer (SML). The NICs produced by the original slab model are significantly larger than the observations, indicating other important processes operating over the area. We followed Welle (1982) and Jing et al. (2017) and used a modified slab model in this study by including contributions from the background geostrophic currents. Using the surface geostrophic currents inferred from the satellite sea level data and assuming the geostrophic currents in the SML is vertically uniform, we found that the modified slab model performs significantly better than the original slab model in reproducing the observed NICs at two ADCP mooring sites in the nSCS. Examinations of observations and numerical results produced by the modified and original slab models revealed the occurrence of the energy exchange between the mesoscale eddies and the NICs. Based on the energy budget analysis for NICs during the observational period, the difference of the near-inertial wind power input between the original slab model and the modified slab model is the same order as the energy transfer rate (Eq. 25). This also indicates the importance of the near-inertial energy transfer induced by the interaction between mesoscale eddies and NICs in the SML during the observational period.

The modified slab model and original slab model were then used to examine sensitivity to winds and eddy parameters with idealized mesoscale eddies under cyclonic winds. Both cyclonic and anticyclonic mesoscale eddies were considered, using the universal eddy structure suggested by Zhang et al. (2013). One of our major findings is that anticyclone eddies can transfer more kinetic energy to NICs than cyclonic eddies. Idealized experiments show that induced NICs speed in anticyclonic eddies can reach over 6 times the speed in cyclonic eddies. We also found that the energy transfer rate is related to the Okubo-Weiss parameter. When the Okubo-Weiss parameter is positive, the energy transfer rate is elevated with the larger Okubo-Weiss parameter.

Analyses of model results in 196 numerical experiments using the modified slab model demonstrated that there exists bidirectional energy transfer between mesoscale eddies and NICs. The direction of the energy transfer is primarily from mesoscale eddies to NICs. When the cyclonic winds rotate at frequencies of $0.25f$, $0.75f$ and $1.25f$ and the anticyclonic winds rotate at frequencies of $-1.1f$, $-0.9f$ and $-0.75f$, the direction of the energy transfer is negative in the cyclonic eddy, that is, from NICs to cyclonic eddies. Under anticyclonically rotating winds at frequencies of $-1.5f$, $-1.25f$, $-1.1f$ and $-0.9f$, the negative energy transfer also occurs in the anticyclonic eddy. The NICs transferred from mesoscale eddies are stronger for higher wind



speeds, faster translational speeds and stronger strengths of mesoscale eddies. When the wind stress increases by a factor of $n$ times, the amplitudes of the converted NICs are also multiplied by $n$ times. The NICs transferred in mesoscale eddies by the interactions between mesoscale eddies and NICs are stronger for higher translational speeds of anticyclonic eddies. At the translational speeds of 11 cm s$^{-1}$, the ratios of the amplitudes of the converted NICs by anticyclonic eddies to that transferred by cyclonic eddies reach maximum values.

For analytical considerations, the modified slab model was transferred from the time domain to the frequency domain using the Fourier transform. Using Parseval's theorem, we derived the time-mean value of the induced NICs. The analytical expression was used to demonstrate that, for the negative relative vorticity, i.e., such as within an anticyclonic eddy, the transferred NICs are larger in an anticyclonic eddy than a cyclonic eddy. The analytical solution under the cyclonic winds also demonstrated that the NICs transferred by mesoscale eddies increase linearly with the wind stress. These analytical results are consistent with the results produced by the modified and original slab models.

We also conducted the same set of numerical experiments using steady winds in both constant speeds and direction and model results in the steady winds are not presented here due to the page limit. Our main findings on the energy transfer between mesoscale eddies and NICs in these experiments with the steady winds are the same as the results using the rotating winds.

This study suggests that there is bidirectional energy transfer between mesoscale eddies and NICs in the SML, of which the mechanism and influence factors are further explored by idealized simulations. Our findings can further contribute to the understanding of the energy budget in the global ocean and the ocean response to the climate change. In order to examine major physical processes affecting the NICs generated by the mesoscale eddies and quantify their influence on turbulent mixing in the deeper ocean, further studies are needed using a three-dimensional ocean circulation model.

**Data availability**

All the data can be obtained by contacting the authors.



**Competing interests**

The authors declare that they have no conflict of interest.

**Acknowledgments**

This study is supported by funds from the Guangdong Basic and Applied Basic Research
Foundation (2022B1515130006), the National Natural Science Foundation of China (91958203)
and Shenzhen Science and Technology Innovation Committee (WDZC20200819105831001). We
thank the editor and reviewers for their useful and constructive comments.

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
