# Peer review of "Parameter Sensitivity Study of Energy Transfer Between Mesoscale Eddies and Wind-Induced Near-Inertial Oscillations"

_EGUsphere, 2024_

## Referee Comment (RC2)

General comments on Figures : Please consider improving the readiness of your figures (add grid, no need for bold labels, font sans serif)

Major comments :
- There is no description of the NIW trapping mechanisms in anticyclones, nor introduction of the concept of effective Coriolis frequency. This could help interpreting some of the results (eg Fig 6), as well as simplifying some of the analytical equations of the slab model [as done by Jing et al. (2017)]. Vorticity estimates should be normalised by f in order to understand better how mesoscale and NIC interact.
- l329 The change in sign of vorticity does not seem to be caused by a velocity reversal (fig 4), but rather the relative importance of the two terms of vorticity in cylindrical coordinates : dv/dr and v/r. Would it be interesting to simulate stations in the opposite vorticity ring around the eddy core?
- l373 and for all the diagnostics : why considering the sum of the 9 simulated stations and not the average and spread ?
- Figure 6 : What is happening at -f? Since it does not seem to be part of the sensitivity test (l417). How does the effective Coriolis frequency influence the results? Would it be possible to run more cases in order to have a better resolution of the peaks around +/-f? Section 5.3 is very descriptive and does not provide any mechanisms to explain the important discrepancies between cyclonic and anticyclonic winds.
- l491/500 Can you be sure that the parameter alpha is peaking at 11cm/s by providing a single point for larger translation speed? How does it compare with the maximum rotation speed of the eddy?
- What is the relationship between epsilon and vorticity and strain separately? It would be useful to color the marker with their experiment letter.

Specific minor comments :
- title : Note sure Parameter is needed in the title
- l34 please check the chronological order of citation and make it consistent.
- l38-52 : The description of NIW trapping in mesoscale eddies should be expanded here [eg Fer et al. (2018), Lelong et al. (2020)], as well as the definition of the effective frequency.
- l90 providing the depth of the moorings could be useful.
- L94 Isn't ERA5 resolution 0.25° ?
- ECMWF being an distributor of the geostrophic currents of Copernicus, I am not sure it is useful to mention it here.
- l103 0.125 sigma units is not really a classical threshold for MLD calculation.
- Figure 1 : What is the mean circulation of the area ? For the nonspecialist reader, could you add arrow of the main circulation features of the study area, or a field of mean dynamic topography. Lon/lat ratio seems to be equal rather than respecting, for instance, a flat projection.
- 3.2 : For how long do you integrate the slab model ? Do you perform your analysis on a steady state ?
- l127 under nearly steady winds
- l133 8 days-1 seems rather arbitrary. How does it compare with other studies ? (same comment in line 239)
- l135 ERA5's 10-m winds
- l166 Why is the smoothing necessary ?
- l167 NICs amplitude ?
- l178 There is no description in the method section of how NICs are calculated with observations.
- l190 I found the date number axis confusing… especially because you also describe seasonal cycle afterward.
- l206 How does vorticity compare with f ?
- l211 You could look at the atlas of mesoscale eddies (eg https://www.aviso.altimetry.fr/en/data/products/value-added-products/global-mesoscale-eddy-trajectory-product/meta3-2-dt.html) to describe eddies passing by the mooring.
- Figure 3 : The days chosen are so far apart that it becomes difficult to

follow structures in the maps. Please highlight the trajectory of eddies you describe.
- l280 The frequency band is different than the one of line 166.
- l281 No window size provided.
- l320 What is the Rossby number of the eddy considered here ? How does it compare with the observed eddy ?
- 5.1 and 5.2 are more method sections than results.
- l377+l385 : increases linearly with cyclonic wind stress (ie quadratic response to wind intensity)
- l403-404 : « The energy generation by the Okubo-Weiss parameter » This sentence is not supported by a Figure. This is the first time the Okubo Weiss parameter is mentioned in the result section.
- In line 402, you mention an increase at the eddy edge. Wouldn't it be interesting to have more stations in the rim of the idealised eddy?
- Why is the response in Fig 5b not symmetric regarding to the eddy center?
- Figure 5b : distance to center normalised by eddy radius could be more useful x-axis station number.
- l505 and other caption: it should be mentioned in the method section that the eddy considered move westward.
- Are the Dirac functions necessary in equation 32/33/34/35?
- l615 : value with positive relative vorticity

References :
- Fer, I., Bosse, A., Ferron, B., & Bouruet-Aubertot, P. (2018). The dissipation of kinetic energy in the Lofoten Basin Eddy. *Journal of Physical Oceanography*, *48*(6), 1299-1316.
- Jing, Z., Wu, L., & Ma, X. (2017). Energy exchange between the mesoscale oceanic eddies and wind-forced near-inertial oscillations. *Journal of Physical Oceanography*, *47*(3), 721-733.
- Lelong, M. P., Cuypers, Y., & Bouruet-Aubertot, P. (2020). Near-inertial energy propagation inside a Mediterranean anticyclonic eddy. *Journal of Physical Oceanography*, *50*(8), 2271-2288.

---

## Author Comment (AC1)

The better performance of the modified slab model against observations. How robust is this result? The selection of the damping coefficient is kind of arbitrary. If a larger damping coefficient were used, the original slab model might perform better while the modified slab model could under-predict the magnitude of near-inertial currents.

Thanks for your suggestions. We have addressed this point with corresponding explanations on lines 252-255 in the revised manuscript.

We agree with you that the different damping coefficients can change the results of near-inertial currents (NICs) produced by the modified and original slab models. To verify the stability of the result, we followed your suggestion and reran the model using three other larger damping coefficients ($r^{-1}$ = 5 days, 6 days and 7 days) and also the annually average mixed layer depth. We found that the NICs patterns are not very sensitive to the different damping coefficients and the different MLDs (Figure R1). On the other hand, previous studies suggest that $r$ is much less than f, therefore the modeled NICs can be not sensitive to the value of $r$ (Alford, 2001).

[Figure]

**Figure R1.** Speeds (m/s) of simulated NICs in the surface mixed layer at stations S2 ((a) and (c)) and S3 ((b) and (d)). Red, black, blue and green dashed lines denote results calculated using the linear slab model applied monthly average MLD, with $r^{-1}$ = 5 days, 6 days, 7 days and 8 days) respectively. Red, black, blue and green solid lines denote results calculated using the modified slab model applied monthly average MLD, with four different values of r respectively. Purple dashed and solid lines respectively denote results calculated using the linear and modified slab model applied annually average MLD, with $r^{-1}$ = 8 days. Yearday is the day relative to 00:00:00 (GMT) on 1 January 2016.

The difference in NIC speed between the modified and original models is attributed to energy transfer between eddies and NICs. However, this difference could also result from the difference in the generation of wind-induced NICs in cyclonic eddies and anticyclonic eddies, rather than energy transfer between eddies and NICs.

Thanks for your suggestions. We fully agree with you that the generation of wind-induced NICs in cyclonic eddies and anticyclonic eddies can be different.

The relative motion between the wind and the current can modulate the wind power input in the presence of anticyclonic and cyclonic eddies. As a result, different wind-generated NICs may be induced in anticyclonic and cyclonic eddies under the same wind conditions.

To determine the difference in the speeds of wind-induced NICs within the anticyclonic eddy and cyclonic eddy, we conducted an additional experiment as you suggested. In this experiment, the original slab model is used to simulate the NICs in the anticyclonic eddy and cyclonic eddy under the same wind conditions, and we use the relative wind speed to calculate the wind stress $\tau$, which is defined as

$$\tau = \rho_0 c_d \left|\overrightarrow{U_{wind}} - \overrightarrow{U_{current}}\right|\left(\overrightarrow{U_{wind}} - \overrightarrow{U_{current}}\right)$$

where $\rho_0$ is the air density, $c_d$ is the drag coefficient, $U_{wind}$ is the wind speed at 10-m height and $U_{current}$ is the current speed. Other experiment settings are the same as ExpA3.

[Figure]

**Figure R2.** Averaged speeds of wind-induced NICs in the anticyclonic eddy and cyclonic eddy. Numbers on the horizontal axis denote nine fixed locations P1 to P9. The wind rotates cyclonically at the inertial frequency.

We demonstrate that the amplitude of the NICs generated in the cyclonic and

anticyclonic eddy is almost the same under the same wind conditions (Figure R2). Therefore, the difference in NIC speed between the modified and original models is induced by the energy conversion between mesoscale eddies and NICs.

The direction of energy transfer between eddies and NICs appears to depend on the wind rotation frequency. Please explain why, for some rotation frequencies, the energy transfer occurs from eddies to NICs, while for others, it shifts from NICs to eddies.

Thanks for your suggestions. We have made some modifications accordingly in Section 5.3 in the revised manuscript.

The energy transfer between mesoscale eddies and NICs is bidirectional, but overall, it is dominated by the positive energy transfer (i.e., energy is transferred from mesoscale eddies to NIWs). When the winds rotate anticyclonically within the frequency range of $-1.15f$ to $-0.75f$, the energy transfer between NICs and mesoscale eddies is negative and relatively strong. These frequencies are close to the inertial frequency of $-f$, which can induce the resonance and generate significant NICs. Therefore, the large NICs provide a strong energy source for reverse energy conversion, and the near-inertial kinetic energy can be reabsorbed into the background mesoscale eddies to help reconstruct the geostrophic balance.

Can you explain why alpha has a maximum value at the eddy translational speed of 11 cm/s? What is special about this translational speed?

Thanks for your valuable comments. We have added some explanations on lines 490-500 and 516-521 in the revised manuscript.

For an anticyclonic eddy with $|SLA_c|$=0.64 m, the sum of average speeds of the transferred NICs at the nine locations ($\sum NICs\_U_{AE}$) increases linearly from about 0.133 m/s to 0.148 m/s as the translational speed increases from 4 cm/s to 11 cm/s. The increase of the translational speed enhances the total kinetic energy of mesoscale eddies, which can provide a larger energy source and be more beneficial for the conversion of NICs. It should be noted that the change in the eddy kinetic energy caused by the different translational speeds is relatively small in comparison with the total eddy energy determined by the mesoscale eddy strength. Therefore, the change in the amplitude of the transferred NICs is relatively small. After the translational speeds reach 11 cm/s, the values of $\sum NICs\_U_{AE}$ are almost the same with that at the translational speed of 11 cm/s.

For a cyclonic eddy with $|SLA_c|$=0.64 m, the sum of average speeds of the transferred NICs at the nine locations ($\sum NICs\_U_{CE}$) are all about 0.066 m/s as the translational speed increases from 4 cm/s to 11 cm/s. However, after the translational speeds reach 11 cm/s, the values of $\sum NICs\_U_{CE}$ are larger with faster translational speeds.

Therefore, the alpha has a maximum value at the eddy translational speed of 11 cm/s. This indicates that the anticyclonic eddy transfers much more near-inertial energy than the cyclonic eddy does, particular at the translational speed of 11 cm/s.

You talked quite a bit about the importance of strain and OW parameter, but your theoretical analysis seems to say that only the relative vorticity matters?

Thanks for your valuable suggestions. We have added some explanations on lines 611-614.

Jing et al. (2017) demonstrated that the strain is responsible for the permanent energy transfer from mesoscale eddies to wind-forced NICs, because the energy transfer efficiency is always zero in absence of the strain. However, in the presence of the strain, the relative vorticity can have an influence on the energy transfer by modifying the effective Coriolis frequency. It means that both strain and relative vorticity have an impact on the energy transfer between the mesoscale eddies and NICs. The strain determines whether the energy transfer occurs, and the relative vorticity changes the magnitude of energy transfer efficiency.

In our numerical experiments, the strain of the ideal mesoscale eddy is nonzero, which means the energy transfer between NICs and mesoscale eddies occurs. Our research aims to explore the sensitivity of the energy transfer between NICs and mesoscale eddies to the wind speed, the translation speed of the mesoscale eddy, the strength of the mesoscale eddy, the wind rotation frequency and the relative vorticity in the presence of the strain. Meanwhile, we also try to better understanding of the role of the relative vorticity in the background flow for anticyclonic eddies to be significantly more efficient than the cyclonic eddies in transferring their kinetic energy to NICs. Therefore, in the theoretical analysis, we mainly explore the influence of relative vorticity on the energy conversion to verify with the numerical experiments.

It feels that your theoretical analysis is not really on energy transfer between eddies and NICs, but on the impact of relative vorticity on the generation of NICs.

Thanks for your suggestions. We have added some explanations on lines 618-621 in the revised manuscript.

It is correct that theoretical analysis is mainly about the impact of relative vorticity on the generation of NICs under steady winds, but this generation of NICs is induced by energy transfer between eddies and NICs instead of changing winds.

Without any background mesoscale eddy, the modified slab model is the same as the original slab model. The simulated NICs by the original slab model represent the near-inertial energy generated directly by the wind forcing. Since the same wind forcing is used in both the original and modified slab models, the differences in the results produced by the modified and original slab model represent the amplitudes of NICs transferred by interactions between mesoscale eddies and NICs in the mesoscale eddies after removing the generation of wind-induced NICs (i.e., the NICs produced by the original slab model).

Under the steady winds, the NICs produced by the original slab model is 0, therefore the NICs produced by the modified slab model directly represent the energy converted between mesoscale eddies and NICs.

Some minor comments:

Any validation of ERA5 winds at mooring locations?

Thanks for your suggestions. We have made some additions on lines 103-105 in the revised manuscript. The wind speed data obtained from ERA5 is widely used in the previous researches on near-inertial motions in the Northwestern Pacific (Yuan et al., 2024; Chen et al., 2023; Chen et al., 2024), which indicates that it has good applicability. Therefore, the validation of ERA5 is not conducted in this study.

Line 166. Why is smoothing required?

Thanks for your comments. The smoothing (running window) is designed to more clearly illustrate the characteristics of the near-inertia period of the currents, especially by eliminating some high-frequency noise signals.

Line 209-211. I don't see two separate weak cyclonic eddies to the east and south in the SLA map.

Accepted and revised.

Section 5.1. The eddy vertical structure is irrelevant here, since only the eddy surface geostrophic velocity is used in the numerical experiments.

Thanks for your suggestions. We fully agree with you that the eddy vertical structure is irrelevant here. Considering that $H(z)$ needs to be used to verify the rationality of the structure of the idealized mesoscale eddy, the vertical structure function was presented here.

Line 590. "Consider steady wind forcing..." There is still wind forcing with frequency omega on RHS of (27).

Thanks for your valuable comments. The omega is generated by the Fourier transform of the time-varying velocity components $(u, v)$. Meanwhile, as the wind forcing is steady, Dirac Delta function is used here.

References

Alford, M. H.: Internal swell generation: The spatial distribution of energy flux from the wind to mixed layer near-inertial motions, J. Phys. Oceanogr., 31, 2359–2368, https://doi.org/10.1175/1520-0485(2001)031,2359:ISGTSD.2.0.CO;2, 2001.

Chen, Z., Chen, Z., Yu, F., Qiang R., Liu, X., Nan, F., Wang, J., Si, G., & Hu, Y. Deep propagation of wind-generated near-inertial waves in the Northern South China Sea. Deep Sea Res. Part I., 204(Feb.):104226.1-104226.11, https://doi.org/10.1016/j.dsr.2023.104226, 2024.

Chen, Z., Yu, F., Chen, Z., Wang, J., Nan, F., Qiang R., Hu, Y, Cao, A., & Zheng, T.: Downward Propagation and Trapping of Near-Inertial Waves by a Westward-Moving Anticyclonic Eddy in the Subtropical Northwestern Pacific Ocean, J. Phys. Oceanogr., 53(9):2105-2120, https://doi.org/10.1175/jpo-d-22-0226.1, 2023.

Yuan, S., Yan, X., Zhang, L., Pang, C., & Hu, D.: Observation of near-inertial waves induced by typhoon Lan in the Northwestern Pacific: Characteristics, energy fluxes and impact on diapycnal mixing, J. Geophys. Res.-Oceans, 129, e2023JC020187, https://doi.org/10.1029/2023JC020187, 2024.

---

## Author Comment (AC2)

**Responses to Reviewer 2:**

General comments on Figures: Please consider improving the readiness of your figures (add grid, no need for bold labels, font sans serif)

Thanks for your valuable suggestions. All Figures have been modified accordingly.

Major comments:

- There is no description of the NIW trapping mechanisms in anticyclones, nor introduction of the concept of effective Coriolis frequency. This could help interpreting some of the results (eg Fig 6), as well as simplifying some of the analytical equations of the slab model [as done by Jing et al. (2017)]. Vorticity estimates should be normalized by $f$ in order to understand better how mesoscale and NIC interact.

Thanks for your suggestions. The description of NIW trapping mechanisms have been added on lines 38-55 in the revised manuscript. The concept of effective Coriolis frequency has been added on lines 125-127 in the revised manuscript. Vorticity estimates have been modified accordingly in the revised manuscript.

- l329 The change in sign of vorticity does not seem to be caused by a velocity reversal (fig 4), but rather the relative importance of the two terms of vorticity in cylindrical coordinates: dv/dr and v/r. Would it be interesting to simulate stations in the opposite vorticity ring around the eddy core?

Thanks for your valuable suggestions. We have revised the description regarding the change in vorticity signs on lines 346-349 and added more explanations about the selection of the positions on lines 422-423 in the revised manuscript.

Following your suggestions, we select six new positions in the opposite vorticity ring region (P1, P2, P3, P13, P14 and P15) and conduct five numerical experiments with different wind speeds. The other settings of these experiments are the same as those in ExpA1-5. The fifteen positions P1-P15 are radially outward from the center to the edge (Figure R1c).

The distance from the eddy center to P1 (P15) to is $1.61R_0$, to P2 (P14) is $1.38R_0$, to P3 (P13) is $1.15R_0$, to P4 (P12) is $0.92R_0$, to P5 (P11) is $0.69R_0$, to P6 (P10) is $0.46R_0$, and to P7 (P9) is $0.23R_0$.

The northernmost location is P1 and the southernmost location is P15. The locations P7-P9 are located near the eddy center, the locations P4-P6 and P10-P12 are located at the eddy edge, and the locations P1-P3 and P13-P15 are located at the opposite vorticity ring.

[Figure]

**Figure R1.** Distributions of (a) zonal (u) and (b) meridional (v) components (m s[-1]) of currents and (c) relative vorticity (s[-1]) for an idealized anticyclonic eddy with of radius of 120 km. Six fixed locations (P1, P2, P3, P13, P14 and P15) denoted by red asterisks along the y-axis in (c) are additional positions.

After adding these six new stations, the sum of averaged speeds of NICs converted from the mesoscale eddy at the fifteen locations increases linearly with the cyclonic wind stress (Figure R2a), which is consistent with the previous results conclusion without these six new stations.

We take the example of anticyclonic eddies to illustrate the variation of NICs transferred in the different positions (Figure R2b). Within the ring region, the transferred near-inertial energy gradually decreased. As the vorticity and strain of the mesoscale eddy are strong within the eddy core but relatively weak in the ring region, and the addition of the new positions does not affect the main conclusions, we selected the eddy core as the main research region in this study.

[Figure]

**Figure R2.** (a) Sum of averaged speeds of transferred NICs at 15 fixed locations P1-P15 as a function of the wind speeds in the anticyclonic eddy (orange line) and cyclonic eddy (blue line), respectively. (b) Averaged speeds of transferred NICs as a function of the wind speeds in the anticyclonic eddy. The black, red, gray, yellow and purple line indicate respectively the wind speed of 5 m s[-1], 10 m s[-1], 13 m s[-1], 15 m s[-1] and 20 m s[-

[1]. Numbers on the horizontal axis in (b) denote nine fixed locations P1 to P15. The wind rotates cyclonically at the inertial frequency. Mesoscale eddies move westward at the translational speed of 8 cm s$^{-1}$ and $|SLA_c|$= 0.64 m.

- l373 and for all the diagnostics: why considering the sum of the 9 simulated stations and not the average and spread?

Thanks for your suggestions. We have made some additions on lines 390-392 to explain it and added further analysis regarding the spread of transferred NICs on lines 419-423 and 520-521 in the revised manuscript.

The conclusions drawn from the average and the sum of these 9 points are consistent. The magnitude of the sum is larger than that of the average, which makes the presentation of the results clearer and more intuitive. Therefore, we have demonstrated the sum of the 9 simulated positions.

Since this study primarily focuses on the sensitivity of total energy conversion in mesoscale eddies to the wind speed, the translation speed of the mesoscale eddy, the strength of the mesoscale eddy, the wind rotation frequency and the relative vorticity, the spread of the 9 simulated stations is simply analyzed. The spatial differences across the mesoscale eddy will be further explored in future research.

- Figure 6: What is happening at -f? Since it does not seem to be part of the sensitivity test (l417). How does the effective Coriolis frequency influence the results? Would it be possible to run more cases in order to have a better resolution of the peaks around +/-f? Section 5.3 is very descriptive and does not provide any mechanisms to explain the important discrepancies between cyclonic and anticyclonic winds.

Thanks for your suggestions. More explanations about the mechanisms to explain the differences between cyclonic and anticyclonic winds are added accordingly in Section 5.3 in the revised manuscript.

When the wind rotation frequency is equal to the inertial frequency, resonance is induced, which results in pronounced near-inertial currents (NICs).

In the presence of the strain, the relative vorticity has an influence on the energy transfer by modifying the effective Coriolis frequency. When the effective Coriolis frequency is smaller than the inertial frequency, the energy transfer efficiency is mostly large, while the opposite case leads to lower energy transfer efficiency.

Following your suggestions, we have added 15 experiments on lines 455-476 in the revised manuscript. In the previously manuscript, 13 numerical experiments using different wind rotation frequencies ($-1.5f$, $-1.25f$, $-1.1f$, $-0.9f$, $-0.75f$, $-0.5f$, $-0.25f$, $0.25f$, $0.5f$, $0.75f$, $f$, $1.25f$ and $1.5f$, where $f$ is the inertial frequency) are conducted (Figure R3). To have a better resolution of the peaks around the inertial frequency, we conducted 28 numerical experiments using different wind rotation frequencies ($-1.5f$, $-1.25f$, $-1.2f$, $-1.15f$, $-1.1f$, $-1.05f$, $-f$, $-0.95f$, $-0.9f$, $-0.85f$, $-0.8f$, $-0.75f$, $-0.5f$, $-0.25f$, $0.25f$, $0.5f$, $0.75f$, $0.8f$, $0.85f$, $0.9f$, $0.95f$, $f$, $1.05f$, $1.1f$, $1.15f$, $1.2f$, $1.25f$ and $1.5f$). Other experiment settings are the same as the previous cases.

The overall conclusions remain consistent with previous results, with a significant near-inertial energy peak value occurring at the inertial frequency -$f$.

[Figure]

**Figure R3.** Sum of averaged speeds of transferred NICs at 9 fixed locations P1-P9 as a function of rotation frequencies of (a) anticyclonically rotating winds and (b) cyclonically rotating winds. The orange and blue line indicate respectively the anticyclonic eddy and cyclonic eddy. The wind rotation frequencies are normalized by the inertial frequency $f$.

- l491/500 Can you be sure that the parameter alpha is peaking at 11cm/s by providing a single point for larger translation speed? How does it compare with the maximum rotation speed of the eddy?

Thanks for your valuable comments. We have made some additions on lines 496-497 in the revised manuscript.

The maximum rotation speed of the eddy in this study is about 1.62 m/s, which is approximately fifteen times the translation speed of 11 cm/s.

To ensure the parameter alpha is peaking at 11cm/s, we conducted new experiments using faster translational speeds (13cm/s, 14cm/s, 15cm/s and 16cm/s). We take the mesoscale eddy with the $|SLA_c|$ of 0.13 m as an example (Figure R4). In these four experiments, the speed of cyclonically rotating winds at the inertial frequency is set to 13 m/s. The mesoscale eddy moves westward.

When the translation speed is less than 11 cm/s, the value of parameter alpha exhibits an increasing trend. However, when the translation speed exceeds 11 cm/s, the value of parameter alpha shows a decreasing trend. Considering the observations on the translation speeds of mesoscale eddies in the northern South China Sea, the range of translation speeds is from 4 cm/s to 12 cm/s in this study. Therefore, the parameter alpha is peaking at 11cm/s within this experimental interval.

[Figure]

**Figure R4.** The $\alpha$ value as a function of the eddy translational speed. The speed of the cyclonic wind is 13 m s$^{-1}$, and the wind rotates at the inertial frequency. Mesoscale eddies move westward and $|SLA_c| = 0.64$ m.

- What is the relationship between epsilon and vorticity and strain separately? It would be useful to color the marker with their experiment letter.

Thanks for your suggestions. We have made some additions on lines 599-604, and we have colored the marker in the Figure 10 in the revised manuscript.

Following your suggestions, we have analyzed the relation between the energy transfer rate ($\varepsilon$) and normal strain and shear strain, separately. When the normal strain (shear strain) is negative, the energy transfer rate shows a decreasing trend with the increase of the Okubo-Weiss parameter decreases. However, when the normal strain (shear strain) is positive, the energy transfer rate is elevated as the normal strain (shear strain) increases (Figure R5 and R6).

[Figure]

**Figure R5.** Scatterplot between the energy transfer rate and the normal strain. Purple, blue and red dots respectively represent ExpA3, ExpC1-9 and ExpD1-16. The left (right) black line is the linear fitting line when the Okubo-Weiss parameters are negative (positive).

[Figure]

**Figure R6.** Scatterplot between the energy transfer rate and the shear strain. Purple, blue and red dots respectively represent ExpA3, ExpC1-9 and ExpD1-16. The left (right) black line is the linear fitting line when the Okubo-Weiss parameters are negative (positive).

There is no significant linear correlation between the energy transfer rate and relative vorticity. However, based on the sensitivity experiments conducted in this study, we have demonstrated anticyclonic eddies generally convert more near-inertial energy than cyclonic eddies under the same conditions.

Specific minor comments:

- title: Note sure Parameter is needed in the title

Accepted and revised.

- l34 please check the chronological order of citation and make it consistent.

Accepted and revised. We have checked and sorted the references here alphabetically by the first letter of the authors' last names to ensure consistency with the reference citation order used elsewhere in the revised manuscript.

- l38-52: The description of NIW trapping in mesoscale eddies should be expanded here [eg Fer et al. (2018), Lelong et al. (2020)], as well as the definition of the effective frequency.

Thanks for your suggestions. We have expanded the description of the trapping mechanism accordingly on lines 38-55 in the revised manuscript, and we have added

the definition of the effective frequency on lines 125-127.

- l90 providing the depth of the moorings could be useful.

Thanks for your suggestions. We have made some additions on line 89 and line 92 in the revised manuscript.

- L94 Isn't ERA5 resolution 0.25°?

Accepted and revised.

- ECMWF being a distributor of the geostrophic currents of Copernicus, I am not sure it is useful to mention it here.

Accepted and revised. We have deleted the description of ECMWF here.

- l103 0.125 sigma units is not really a classical threshold for MLD calculation.

Thanks for your valuable comments. We will select datasets of MLD defined by other threshold in future research.

- Figure 1: What is the mean circulation of the area? For the nonspecialist reader, could you add arrow of the main circulation features of the study area, or a field of mean dynamic topography. Lon/lat ratio seems to be equal rather than respecting, for instance, a flat projection.

Thanks for your suggestions. Figure 1 has been modified accordingly.

- 3.2: For how long do you integrate the slab model? Do you perform your analysis on a steady state?

Thanks for your comments. We have added more explanations on lines 341-342 in the revised manuscript. We integrate the slab model for more than 3000 hours. After the wind-driven currents reach a steady state, we let a mesoscale eddy propagate westward to reach the area of interest (positions P1-P9), at which the model results are band-passed to get the NICs under the cyclonic winds.

- l127 under nearly steady winds

Accepted and revised.

- l133 8 days-1 seems rather arbitrary. How does it compare with other studies? (same comment in line 239)

Thanks for your suggestions. We have addressed this point with corresponding explanations on lines 252-255 in the revised manuscript.

The decay timescale ($r^{-1}$) of 2-20 days has been reported in previous studies (D'Asaro, 1985; Park et al., 2009; Plueddemann & Farrar, 2006).

We agree with you that the different damping coefficients can influence the results of near-inertial currents (NICs) produced by the modified and original slab models. Following your suggestions, we reran the model using three other larger damping coefficients ($r^{-1}$ = 5 days, 6 days and 7 days). We found that the NICs patterns are not

very sensitive to the different damping coefficients (Figure R7). On the other hand, previous studies suggest that $r$ is much less than $f$, therefore the modeled NICs is not sensitive to the value of $r$ (Alford, 2001).

[Figure]

Figure R7. Speeds (m/s) of simulated NICs in the surface mixed layer at stations (a) S2 and (b) S3. Red, black, blue and green dashed lines denote results calculated using the linear slab model applied monthly average MLD, with $r^{-1}$ = 5 days, 6 days, 7 days and 8 days) respectively. Yearday is the day relative to 00:00:00 (GMT) on 1 January 2016.

- l135 ERA5's 10-m winds

Accepted and revised.

- l166 Why is the smoothing necessary?

Thanks for your comments. The smoothing (running window) is designed to more clearly illustrate the characteristics of the near-inertia period of the currents, especially by eliminating some high-frequency noise signals.

- l167 NICs amplitude?

Accepted and revised.

- l178 There is no description in the method section of how NICs are calculated with observations.

Thanks for your suggestions. We have made some additions to the calculation of observed NICs on lines 173-174 and 177-179.

- l190 I found the date number axis confusing⋯ especially because you also describe seasonal cycle afterward.

Thanks for your comments. We have made some modifications to the definition of yearday on lines 198-199 and added the yearday values for different months on lines 207-217.

The yearday is defined as the number of days elapsed since 00:00:00 (GMT) on 1 January 2016. For example, 00:00:00 (GMT) on 2 January 2016 is defined as day 1.

- l206 How does vorticity compare with f?

Thanks for your suggestions. The vorticity is approximately $-0.17f$ at mooring S2 and $-0.13f$ at mooring S3. We also have made some additions on lines 220, 225-226 and 230-231 in the revised manuscript.

- l211 You could look at the atlas of mesoscale eddies (eg https://www.aviso.altimetry.fr/en/data/products/value-addedproducts/global-mesoscale-eddy-trajectory-product/meta3-2-dt.html) to describe eddies passing by the mooring.

Accepted and revised. We have described different mesoscale eddies separately and added more information about mesoscale eddies in the revised manuscript.

- Figure 3: The days chosen are so far apart that it becomes difficult to follow structures in the maps. Please highlight the trajectory of eddies you describe.

Thanks for your suggestions. We have made some modifications on lines 218-233 in the revised manuscript. As the relative negative vorticity was strengthened between days 289-300 (283-297) at station S2 (S3), it may be related to anticyclonic eddy. It is necessary to show the SSH (SLA) and relative vorticity contours to determine the presence of anticyclonic eddy near the mooring sites, therefore the SLA and relative vorticity contours on days 295, 316, 368, 435 and 452 (Figure 3) are presented. The mesoscale eddy on days 295 and 316 is the same, and the mesoscale eddy on days 368, 435 and 452 is the same. Therefore, we primarily aim to reveal the presence of mesoscale eddies near the station S2 (S3) in this study.

- l280 The frequency band is different than the one of line 166.

Thank you for your comments. In determining the observed NICs from ADCP currents data, we used the frequency band of 0.85-1.15f mainly to filter out the diurnal tide signals. In the numerical experiments in Section 5, the simulated currents are mainly the Ekman currents and NICs without tides. As a result, we used a wide frequency band of 0.6-1.4f band toto capture all the NICs signals from the model results. It should be noted that the main conclusion is the same if the narrow band 0.85-1.15f is used in Section 5.

- l281 No window size provided.

Accepted and revised.

- l320 What is the Rossby number of the eddy considered here? How does it compare with the observed eddy?

Thanks for your suggestions. We have added the Rossby number on line 335 in the revised manuscript. The core Rossby number of the idealized eddy can reach -0.7. The core Rossby number of the observed eddy is about -0.2, which is large than that of the idealized eddy.

- 5.1 and 5.2 are more method sections than results.

Thanks for your valuable suggestions. We agree with you that Section 5.1 is the method section. Considering that the idealized mesoscale eddy is primarily utilized in the numerical experiments and Section 5 is presented in parallel with the observations (Section 4), the description of the idealized mesoscale eddy has been presented in Section 5 in this study.

- l377+l385: increases linearly with cyclonic wind stress (ie quadratic response to wind

intensity)

Accepted and revised.

- l403-404: « The energy generation by the Okubo-Weiss parameter » This sentence is not supported by a Figure. This is the first time the Okubo Weiss parameter is mentioned in the result section.

Accepted and revised. We have made some additions to the Okubo Weiss parameter on lines 424-426 in the revised manuscript.

- In line 402, you mention an increase at the eddy edge. Wouldn't it be interesting to have more stations in the rim of the idealised eddy?

Thanks for your suggestions. We have made some additions on lines 420-423.

Following your suggestions, we have added more positions in the rim of the idealized eddy and conducted several new experiments (Figure R1). We found that the amplitudes of NICs transferred in the rim are small and decrease outward from the eddy edge. Meanwhile, the main conclusions are consistent with conclusions before the addition of these positions. Therefore, we selected the nine positions (P1-P9) as the area of interest in this study.

- Why is the response in Fig 5b not symmetric regarding to the eddy center?

Thanks for your comments. After the mesoscale eddy moved and interacted with the NICs, the background flows can change, resulting in slightly more near-inertial energy transferred on the southern side of the mesoscale eddy than that on the northern side under the relatively strong wind conditions.

- Figure 5b: distance to center normalised by eddy radius could be more useful x-axis station number.

Thanks for your suggestions. To provide a clearer description of the distance from different positions to the eddy center, we have made direct additions to the first definition of P1-P9 on lines 340-341.

- l505 and other caption: it should be mentioned in the method section that the eddy considered move westward.

Thanks for your suggestions. We have added the movement direction of the mesoscale eddy on lines 332-333 of the method section.

- Are the Dirac functions necessary in equation 32/33/34/35?

Thanks for your suggestions. This function is necessary due to the transformation of the wind forcing from the time domain to the frequency domain.

- l615: value with positive relative vorticity

Accepted and revised.

References:

Alford, M. H.: Internal swell generation: The spatial distribution of energy flux from the wind to mixed layer near-inertial motions, J. Phys. Oceanogr., 31, 2359–2368, https://doi.org/10.1175/1520-0485(2001)031,2359:ISGTSD.2.0.CO;2, 2001.

D'Asaro E. A.: The energy flux from the wind to near-inertial motions in the surface mixed layer, J. Phys. Oceanogr., 15(8), 1043–1059, https://doi.org/10.1175/1520-0485(1985)015<1043:tefftw>2.0.co;2, 1985.

Park, J. J., K. Kim, & R. W. Schmitt: Global distribution of the decay timescale of mixed layer inertial motions observed by satellite-tracked drifters, J. Geophys. Res., 114, C11010, https://doi.org/10.1029/2008JC005216, 2009.

Plueddemann, A., & J. Farrar: Observations and models of the energy flux from the wind to mixed-layer inertial currents, Deep Sea Res. Part II, 53, 5-30, https://doi.org/10.1016/j.dsr2.2005.10.017, 2006.